# Prediction of SARS-CoV-2 transmission dynamics based on population-level cycle threshold values: An epidemic transmission and machine learning modeling study

Afraz Arif Khan[1], Hind Sbihi[1,2], Michael A Irvine[1,2], Agatha N Jassem[1,3], Yayuk Joffres[1,2], Braeden Klaver[1,2], Naveed Janjua[1,2], Aamir Bharmal[2,4], Carmen H Ng[4], Chris D Fjell[1,3], Miguel Imperial[5], Susan Roman[6], Marthe K Charles[3,7], Amanda Wilmer[8], John Galbraith[9], Marc G Romney[3,10], Bonnie Henry[11], Linda MN Hoang[1,3], Mel Krajden[1,3], Catherine A Hogan[1,3]*

[1]British Columbia Centre for Disease Control, Vancouver, Canada; [2]School of Population and Public Health, University of British Columbia, Vancouver, Canada; [3]Department of Pathology and Laboratory Medicine, University of British Columbia, Vancouver, Canada; [4]Office of the Medical Health Officer, Fraser Health, Surrey, Canada; [5]LifeLabs, Surrey, Canada; [6]Fraser Health, Surrey, Canada; [7]Division of Medical Microbiology & Infection Control, Vancouver Coastal Health Authority, Vancouver, Canada; [8]Division of Medical Microbiology, Kelowna General Hospital, Kelowna, Canada; [9]Division of Microbiology and Molecular Diagnostics, Victoria General Hospital, Victoria, Canada; [10]Division of Medical Microbiology and Virology, St. Paul's Hospital, Vancouver, Canada; [11]Ministry of Health, Victoria, Canada

*For correspondence:
catherine.hogan@bccdc.ca

**Abstract** Polymerase chain reaction (PCR) cycle threshold (Ct) values can be used to estimate the viral burden of Severe Acute Respiratory Syndrome Coronavirus type 2 (SARS-CoV-2) and predict population-level epidemic trends. We investigated the use of epidemic transmission modeling and machine learning (ML) based on Ct value distribution for SARS-CoV-2 incidence prediction in British Columbia, Canada, during an Omicron subvariant BA.1-predominant period from November 2021 to January 2022. Using real-world data, we developed an epidemic transmission model that was first validated on outbreak data and subsequently fitted to province-level data to predict incidence. Using simulated data, we developed a ML pipeline including five models to predict the reproductive number as a measure of transmission potential based on Ct value distribution and validated it on out-of-sample province-level data. The epidemic transmission model demonstrated accurate prediction with the real incidence falling within the 95% credible interval of the predicted MCMC chains for both the long-term care facility outbreak and province-level data. The ML models demonstrated good performance with a mean squared error (MSE) lower than 0.17 across all models and improved performance with increasing sample size. The variability of the Ct distribution around the mean was the strongest predictor of the reproductive number. These modeling approaches demonstrated utility for incidence and reproductive number prediction and have potential to complement traditional surveillance in real time to guide public health interventions.

## Editor's evaluation

This useful study presents an application of modelling approaches for estimating SARS-CoV-2 epidemiological dynamics from polymerase chain reaction (PCR) cycle threshold data. The authors provide solid evidence for their claim that their analytical pipeline has potential utility as a complement to analyses of traditional surveillance data. The results will be of interest to mathematical modellers working on communicable diseases with access to PCR data.

## Introduction

Severe Acute Respiratory Syndrome Coronavirus type 2 (SARS-CoV-2) viral burden can be quantitated by polymerase chain reaction (PCR) cycle threshold (Ct) values, which are inversely proportional to the log amount of target viral sequence present in the patient sample. Although this information is frequently available from routine molecular methods for the diagnosis of SARS-CoV-2 infection, clinical results are almost universally reported qualitatively as present or absent due to sources of sampling variability, lack of inter-test standardization, insufficient supporting clinical correlation data, and lack of regulatory approval for purposes other than qualitative reporting, all of which limit interpretation of Ct values for clinical care. Though the use of Ct values to guide individual-level management is not currently routinely recommended (*Infectious Diseases Society of America and Association of Molecular Pathology, 2021*; *American Association for Clinical Chemistry, 2021*), the assessment of aggregated Ct values at a population level may be useful to help assess early epidemiological transmission trends to improve epidemic forecasting (*Walker et al., 2021*; *Phillips et al., 2022*; *Hay et al., 2021*; *Lin et al., 2022*; *Tsang et al., 2021*; *De Arcos-Jiménez et al., 2025*), and parallels the concept of measuring community viral load used for other viruses (*Herbeck and Tanser, 2016*; *Das et al., 2010*; *Jordan et al., 2020*). Accurate projection of epidemic trends is critical to effectively plan public health efforts including healthcare resource allocation. Indeed, an epidemic in the growth phase is more likely to be associated with high viral load burden at a population level; conversely, the decline phase of an epidemic is likely to demonstrate lower viral burden (*Hay et al., 2021*). A modeling approach was previously published to inform epidemic SARS-CoV-2 trajectory based on aggregated Ct value data (*Hay et al., 2021*), and supported the usefulness of population-level Ct value analysis. However, SARS-CoV-2 testing practices globally have evolved substantially during the pandemic, frequently by restricting testing to symptomatic individuals, and additional modeling approaches may provide complementary information. Starting in December 2021 in British Columbia (BC), use of PCR testing was partially restricted in the context of roll-out of rapid antigen tests, limiting understanding of population trends. Complementary tools are needed to estimate incidence. This includes modeling approaches robust to varying testing guidelines, sample selection strategies, and epidemiologic settings. In this study, we investigated the use of epidemic transmission modeling and machine learning (ML) including five models (Lasso, Light Gradient Boosting Machine [LGBM], Extreme Gradient Boosting [XGBoost], Categorical Boosting [CatBoost], Random Forest [RF]), based on Ct value distribution for SARS-CoV-2 incidence prediction in British Columbia, Canada, during an Omicron-predominant period from November 2021 to January 2022.

## Methods

### Study design

Three pandemic phases in BC were considered based on vaccination roll-out and variant of concern (VoC) distribution (*Supplementary file 1 and 2*); however, the current study focused solely on phase 3 to include the largest representation of asymptomatic individuals. This study population represented a heterogeneous mix of vaccinated and unvaccinated individuals and predominantly Omicron variant BA.1. Individuals with PCR-confirmed SARS-CoV-2 infection by nasopharyngeal swab between November 19, 2021, and January 8, 2022, were included, capturing emergence of the Omicron wave in the province. Descriptive analyses of Ct value distribution included the two main specimen type categories: nasopharyngeal swabs and gargles, while modeling analyses focused on nasopharyngeal swabs given the higher diagnostic yield and collection standardization.

## Testing practices and public health measures

COVID-19 testing practices changed over time in BC. From January 2021 onward, testing was prioritized for individuals at increased risk of severe disease or who worked in high-risk settings (*British Columbia Ministry of Health, 2022*; *Coronavirus COVID-19, 2020*). In addition, starting in December 2021, testing of asymptomatic and mildly symptomatic individuals was initiated with the organized roll-out of rapid antigen tests. During the time course of the study, SARS-CoV-2 molecular testing was performed at both the reference public health laboratory (BCCDC PHL) and at first-line laboratories across the province.

## Laboratory data: SARS-CoV-2 diagnostic testing

SARS-CoV-2 diagnostic testing assays based on the *E* gene target were included for this study. The *E* gene was selected as it was the most commonly tested target across the participating laboratory sites. Testing was performed at two main sites and included the BCCDC PHL laboratory-developed test (LDT) (*Hogan et al., 2021*) and the Panther Fusion SARS-CoV-2 assay (Hologic, Malborough, MA) (*Supplementary file 3*). For individuals having undergone repeat SARS-CoV-2 testing within a 1-week period, only the first positive test per person was included.

## Laboratory data: Variant of concern identification

Testing strategies at the BCCDC Public Health Laboratory (PHL) including VoC screening and confirmation by whole genome sequencing (WGS) when applicable changed over the course of the SARS-CoV-2 pandemic as previously described (*Hogan et al., 2021*). From September 2021, owing to increased case burden and limited sequencing capacity, there was a transition from WGS of all samples to a subset positive SARS-CoV-2 samples. This subset comprised targeted surveillance (cases from outbreaks, vaccine escape, reinfection, and travel-related), and representative baseline surveillance. In addition, 100% of positive samples underwent WGS in the first week of each month. Starting November 15, 2021, in the context of the Omicron variant emergence, WGS was resumed for all samples. Owing to the high transmissibility of Omicron and the surge in case load, starting December 21, 2021, there was a transition from full sequencing to sequencing a subset of representative positive samples in addition to priority cases (including outbreaks, long-term care, vaccine escape, travel-related, hospitalization). Full VoC characterization for the province of BC is described separately (*Figure 1—figure supplement 1*).

## Vaccination status

Vaccination status was defined based on the date of vaccine receipt relative to the date of the sample collection included for the study (*Figure 1—figure supplement 2*; *Government of Canada, 2022*). For the primary dose series, all mRNA (Pfizer, Moderna) and viral vector vaccines (AstraZeneca, Janssen) were considered. For the Janssen vaccine only, fully vaccinated status was defined as having received one dose 14 days or more prior to sample collection. For all other vaccines, **Unvaccinated status** was defined as having received no SARS-CoV-2 vaccine, or having received a SARS-CoV-2 vaccine less than 21 days prior to the sample collection date. **Partially vaccinated** status was defined as having received the SARS-CoV-2 vaccine dose 1 greater than or equal to 21 days prior to sample collection, but having received dose 2 less than 14 days prior to the sample collection. **Fully vaccinated** status was defined as greater than or equal to 14 days since the receipt of dose 2, but having received dose 3 less than 14 days prior to the sample collection. Cross-over vaccination was considered in the same category as homologous vaccine schedules.

## Outbreak case study

To validate the models as described further below, analysis was performed using a well-characterized outbreak in a long-term care facility that occurred in BC. This outbreak was selected on the basis of its large-scale asymptomatic testing size, and generalizability of the affected population. Testing was done weekly until no additional cases were identified within 14 days of the last exposure. There were seven rounds of weekly testing at the outbreak facility, all negative residents and staff were tested for each round. Anyone who developed symptoms was also tested. The epidemiologic data and curve describing the outbreak are presented separately (*Figure 3—figure supplement 1*). Analysis was

based on SARS-CoV-2 *E* gene target, or the *ORF1* gene target instead if the *E* gene target result was not available.

## Data sources

Two main data sources were employed for this study: (1) the Provincial Health Laboratory Viewer and Reporter (PLOVER) database which includes the laboratory diagnostic datasets and (2) the Provincial Immunization Registry (PIR) dataset which includes vaccination data. The laboratory datasets house data on SARS-CoV-2 testing (including date of collection, specimen type, diagnostic quantitative PCR gene target results, VoC screening, and SARS-CoV-2 lineage based on WGS), and individual-level epidemiological data (including age, sex, patient, as well as ordering physician health authority). Gene target results include Ct values of the *E* and ORF1 targets. For the outbreak case study, additional data were directly gathered from public health partners as these were not otherwise available through provincial datasets. Data linkages were performed between the laboratory and PIR datasets through a sequential deterministic linkage based on a minimum of three personal identifiers (personal health number, last name with first three digits of first name, and date of birth). These linkages were performed prospectively on a weekly basis, and specimens with unsuccessful linkages were excluded from the study.

## Ethics

This research was approved by the University of British Columbia Research Ethics (H20-0297 BCC19C-COVID-19 Research).

## Models

### Epidemic transmission (SEIR) and machine learning (ML) models

This study compared two different approaches for inference: epidemic transmission modeling to predict incidence, and ML modeling to predict the reproductive number ($R_t$). The first modeling approach was adapted from an existing methodology (*Hay et al., 2021*) and is based on a compartmental SEIR model that captures different stages in individual infections, namely Susceptible, Exposed, Infectious, and Recovered. In brief, this previously published model uses population-level viral load distributions calibrated to known features of SARS-CoV-2 viral load kinetics to estimate the epidemic trajectory from single or multiple cross-sections of positive samples and was initially validated long-term care facility outbreak data in Massachusetts. Using this approach, discrete Ct values are incorporated in the compartmental model over a series of time horizons. Horizons refer to time points across the sample period which draw on the Ct values to search across the Markov chain Monte Carlo (MCMC) chains to predict the incidence of the sample period.

Application of the SEIR model in the current study was performed on real-world data from a long-term care facility outbreak and on simulated data. First, the SEIR model was validated on data from a long-term care facility outbreak that occurred in BC where point prevalence testing was performed at infrequent intervals as described above. For this purpose, the proportion infected at seed time ($I_0$) was fixed to 1/n, where n corresponds to the total population in the outbreak facility, and the horizon was fixed at size three. Second, following validation on the outbreak data, the SEIR model was initially applied to province-level data for all infected individuals, irrespective of symptom status. This analysis yielded poor incidence prediction performance that was likely a result of the biased testing guidance and sample selection. Indeed, current mathematical models that make use of cross-sectional Ct values to infer epidemic trajectories rely on random sampling of the population to accurately predict epidemic trends (*Hay et al., 2021*). The asymptomatic population in our setting represented the best proxy for frequent, non-symptom-based sampling as testing occurred in the context of occupational screening or pre-travel. Thus, the current study SEIR model was subsequently fitted to province-level Ct value distribution data from asymptomatic individuals using a MCMC framework. This uses a modified Metropolis–Hastings algorithm that incorporates discrete Ct values to generate univariate uniform proposals. Here, horizons were set to sizes 5, 6, and 7 based on separability of sample dates and availability of data.

The SEIR model performance was considered accurate if the true incidence fell within the predicted incidence of the 95% credible interval of the MCMC chains. Modifications to the viral kinetics for the SEIR model were applied to the provincial data to account for the specific nature of the Omicron

(BA.1) variant (**Brandal et al., 2021**; **Zeng et al., 2023**). Based on earlier evidence, the initial time ($t_0$) was fixed to 1 day, the incubation time was fixed to 3 days, and the infectious period was fixed to the default value of 4 days (**Hay et al., 2021**; **Brandal et al., 2021**). Fixed here implies that the viral kinetics were made static rather than dynamic by searching for the parameters via the MCMC framework, and these values were fixed due to sparsity of data. In addition, the model searched for $I_0$, and the upper bound was set to 0.1 based on estimated provincial incidence during the timeline of the study.

## Data simulation

We then applied the SEIR model in the current study to simulated data. To simulate infection times and Ct values, this SEIR model was adapted from the virosolver package (**Hay et al., 2021**). All code used in the current study to enable reproduction is provided separately (https://github.com/BCCDC-DSI/Vital-E-paper, copy archived at **Khan, 2026**). The simulation sample period was set at 140 calendar days. Ct values were generated to simulate a sufficiently large random sample of a population and were applied on a sample size of 1000 on a simulated population of 500,000 individuals. Based on this approach, the default viral kinetics including $R_0$ and $I_0$ from the virosolver package were used (**Hay et al., 2021**). We set the default parameters for the simulation of Ct Data including N (population size), sample_days (the number of days to simulate over), and sample_size (the number of people per day). Of note, N/sample_days must be a natural number. After we set up the simulation parameters, we simulate an SEIR model using the default SEIR parameters in the virosolver package (**Supplementary file 4**). This enables us to solve the SEIR model as an ordinary differential equation (ODE). This uses the lsoda package on CRAN that utilizes the specified initial conditions listed in **Supplementary file 4**. Once we have the solved SEIR model, we simulate the Reproductive Number (Rt) [Rt = S*R0], where S represents the susceptible population from the solved SEIR model, and R0 represents the initial value of the reproductive number that is set as default to 2. We then simulate the infection times by using the probability of infection that we obtain from the solved SEIR model's I (number of Infected) parameter. Once we have the infection time, we can create the time since infected variable for each sample in our simulation study, which finally enables us to simulate viral cycle threshold (Ct) values. Here we make use of virosolver's built-in methods that require the time since infected, input parameters, and additional mechanisms to generate a Ct value for each sample. Finally, we join our data to the simulated Rt and synthesize a simulated dataset.

## ML models

The second modeling approach was based on a collection of ML approaches for prediction of the reproductive number on simulated data, including Lasso (**Tibshirani, 1996**), RF, LGBM, XGBM, and CatBoost. A separate real-world data analysis by ML was planned for the study to ensure head-to-head comparison between the models; however, due to the insufficient number of randomly tested samples with which to conduct the study with real-world data, performance was very limited with preliminary analyses and precluded further ML work based on real-world data. Thus, only simulated data were used for ML analysis. To simulate infection times and Ct values, the separate deterministic SEIR model adapted from the virosolver package was used for the machine learning approach. All code used in the current study to enable reproduction is provided separately (https://github.com/BCCDC-DSI/Vital-E-paper, **Khan, 2026**). The simulation sample period was set at 140 calendar days to encompass a typical single SARS-CoV-2 wave. Ct values were generated to simulate a sufficiently large random sample of a population and were applied on sample sizes of 100, 1000, and 10,000 on a simulated population of 500,000 individuals. Using this approach, daily Ct value data are aggregated and incorporated as moments including mean, median, variance, skewness, and kurtosis, rather than incorporated as discrete Ct values. These features were subsequently used to predict the $R_t$ across all ML models. The trained data were generated from a unique simulation file with a fixed random seed and three distinct sample sizes, so three simulated datasets were investigated in this study. Hyperparameter tuning was performed via a grid search of hyperparameters on each model (**Supplementary file 5**). The best-performing model was chosen by identifying the optimal set of hyperparameters for which the mean squared error (MSE) between the true simulated $R_t$ and predicted simulated $R_t$ was minimized on out-of-sample data. SHapley Additive exPlanation (SHAP) analysis was performed for ranking importance of each feature on the prediction of $R_t$ (**Lundberg S. and Lee, 2017**). Based on

these analyses, we were able to produce head-to-head result comparison from the SEIR and the ML models comparison for simulated data, and to produce analysis on real-world data for the SEIR model only.

## Results
### Cohort description
During the study period, a total of 500,914 SARS-CoV-2 tests were performed in BC, of which 70,704 were positive (*Figure 1A and B*). The Omicron (BA.1) variant predominated throughout the period of the study (*Table 1* and *Figure 1—figure supplement 1*). By the end of the study period, a total of 15,494 (21.9%) were unvaccinated, 1605 (2.3%) had received one dose of vaccine, and 49,361 (69.8%) were fully vaccinated (*Table 1*). The *E* gene Ct distribution for the entire province across the study period varied between 20.5 (interquartile range [IQR], 6.8) and 21.6 (IQR 9.1)(*Figure 2A*). The greatest width of the violin plots, corresponding to the highest probability of sampling lower Ct values, coincided with the incidence peaks observed for the same time points for the province (*Figure 2B*).

During the study period, a total of 500,914 SARS-CoV-2 tests were performed in BC, of which 70,704 were positive (*Figure 1A and B*). The Omicron (BA.1) variant predominated throughout the period of the study (*Table 1* and *Figure 1—figure supplement 1*). By the end of the study period, a total of 15,494 (21.9%) were unvaccinated, 1605 (2.3%) had received one dose of vaccine, and 49,361 (69.8%) were fully vaccinated (*Table 1*). The *E* gene Ct distribution for the entire province across the study period varied between 20.5 (interquartile range [IQR], 6.8) and 21.6 (IQR 9.1) (*Figure 2A*). The greatest width of the violin plots, corresponding to the highest probability of sampling lower Ct values, coincided with the incidence peaks observed for the same time points for the province (*Figure 2B*).

### Real-world data: SEIR model outbreak case study validation
This outbreak occurred in a long-term care facility and resulted in a total of 156 individuals (93 residents and 63 staff) infected with SARS-CoV-2 (*Figure 3—figure supplement 1*). Of these infected individuals, 58.1% were asymptomatic in the residents, whereas 9.5% were asymptomatic within the staff. There were 26 (28.0%) deaths in the resident group, and no deaths among the staff. A single horizon with three time points was used for constructing the multiple cross-section SEIR model on the outbreak data. This model showed a peak in incidence on the seventh day of the outbreak, which preceded by 3 days the observed peak at the outbreak facility (*Figure 3A*). The SEIR model demonstrated reasonable prediction with the real incidence falling within the 95% credible interval of the predicted MCMC chains. A violin plot of the posterior samples shows a low outbreak incubation time with a median of 2.6 days and a high initial $R_t$ with a median of 9.5 days (*Figure 3B*). The Ct posterior predictive distribution across the three time points showed an increase in Ct values between days 12–19 as the number of cases waned (*Figure 3C*).

### Real-world data: SEIR model provincial-level
Restricting the provincial-level data to tests conducted on asymptomatic individuals left 429 first positive tests per person for analysis (*Table 1*). In this subgroup, most individuals were in the 19–39-year-old (63.2%) and 40–59-year-old (23.8%) age groups. The Omicron variant accounted for all infections, and most individuals were unvaccinated (82.8%). Of the three horizon sizes tested, the best results were observed with a horizon of size 7 where the real incidence fell within the 95% credible interval of the predicted MCMC chains of the SEIR model (*Figure 4A*). The model posteriors indicated an incidence peak from December 27, 2021, to January 1, 2022, which overlapped with the observed peak of 38 reported cases in the province for this cohort. Similarly, the exponential growth phase coincided with the increase from 8 to 38 reported cases from our cohort from December 20, 2021, to December 27, 2021, and the decline of the incidence coincided with the decline from 17 to 10 cohort cases, from January 1, 2022, to January 5, 2022 (*Figure 4A*). A violin plot of the posterior shows a high initial $R_t$ with a median of 5 days (*Figure 4B*) for the largest horizon. The posterior predictive Ct distribution approximated the observed Ct distribution on the largest time horizon, supporting reasonable incidence projection independent of biases of testing guidance. The Ct posterior predictive distribution

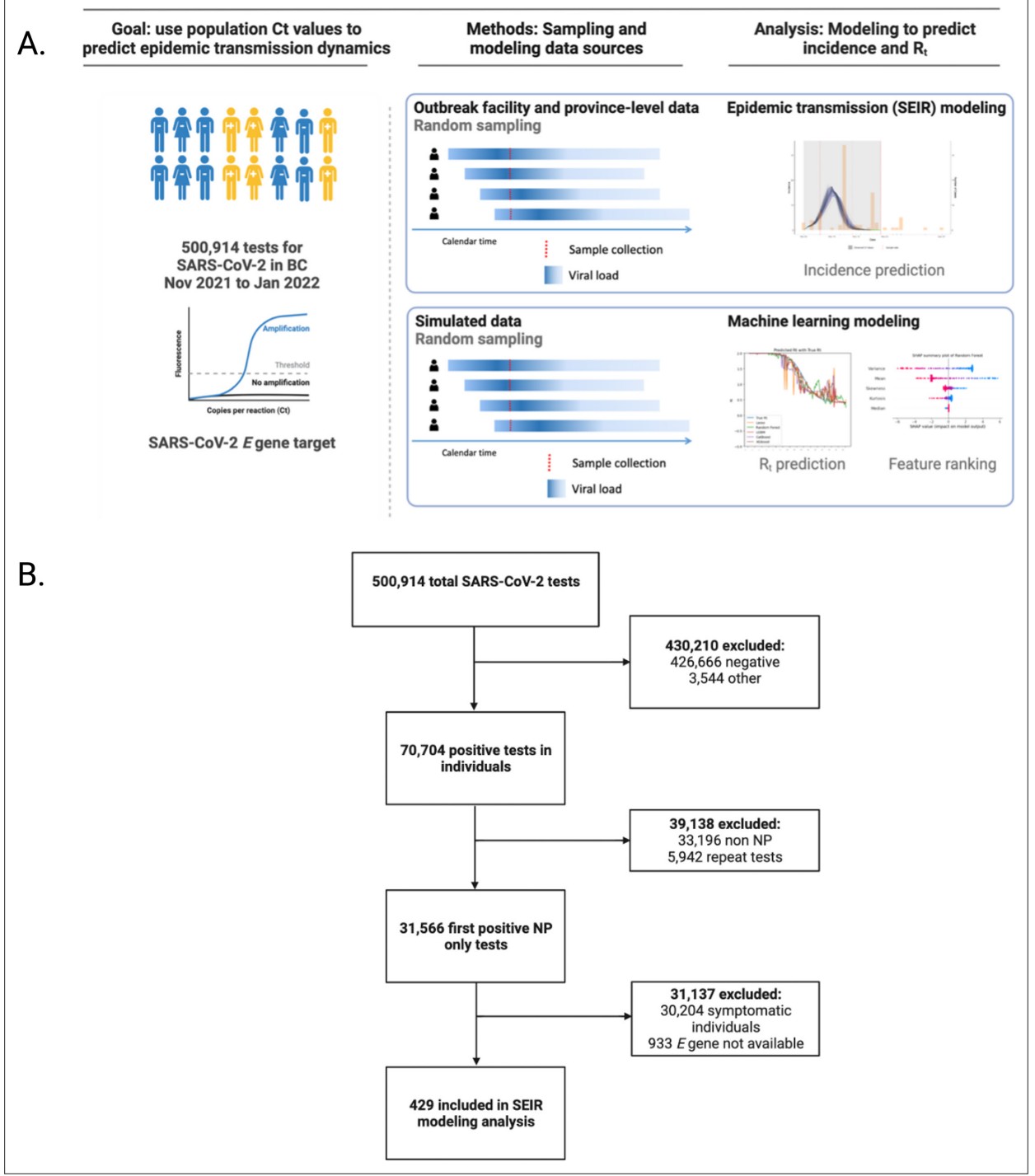

**Figure 1.** Study design. Overall design of the study (**A**) showing the patient population and molecular target of interest for the cycle threshold modeling (first column), sampling and modeling data sources for the study (second column), and modeling approaches (third column). The overall study flowchart (**B**) is also presented and depicts the process that led to selection of 429 tests for the SEIR modeling analysis. BC: British Columbia; E gene: Envelope gene; SARS-CoV-2: Nov: November; Rt: reproductive number; SARS-CoV-2: severe acute respiratory syndrome coronavirus type 2; SEIR: susceptible-exposed-infectious-recovered.

The online version of this article includes the following figure supplement(s) for figure 1:

**Figure supplement 1.** Twenty most prevalent SARS-CoV-2 variant of concern lineages in British Columbia from January 2021 to January 2022.

**Figure supplement 2.** Vaccination status definitions.

**Table 1.** Epidemiological, clinical, and laboratory data of the cohort of asymptomatic individuals tested during the test period of the study.

| Group | Subgroup | Phase 3 (n=500,914) | Subgroup for SEIR analysis (n=429) |
|---|---|---|---|
| Testing* | Positives | 70,704 | 429 |
| | Negatives | 426,666 | 0 |
| | Repeats | 5,942 | 0 |
| | Other | 3,544 | 0 |
| Specimen type | NP | 32,956 | 429 |
| | SG | 37,508 | 0 |
| | Other | 71 | 0 |
| Age (years) | 0–4 | 2,013 | 2 |
| | 5–18 | 8,757 | 23 |
| | 19–39 | 31,497 | 271 |
| | 40–59 | 19,535 | 102 |
| | 60–79 | 7,518 | 29 |
| | ≥80 | 1,376 | 2 |
| | Unknown | 8 | 0 |
| Sex | Female | 37,073 | 206 |
| | Male | 32,733 | 221 |
| | Unknown | 898 | 2 |
| Patient health authority | 1 | 31,490 | 151 |
| | 2 | 9,000 | 4 |
| | 3 | 3,825 | 5 |
| | 4 | 16,394 | 245 |
| | 5 | 9,684 | 8 |
| | Unknown | 311 | 16 |
| Vaccination status | Unvaccinated | 15,494 | 355 |
| | One dose | 1,605 | 4 |
| | Fully vaccinated[†] | 49,361 | 64 |
| | Other | 4,244 | 6 |
| Asymptomatic testing | | 1,548 | 429 |
| No *E* gene result | | 18,583 | 0 |
| VoC lineage | Alpha (B.1.1.7) | 0 | 0 |
| | Beta (B.1.351) | 0 | 0 |
| | Delta (B.1.617.2) | 9,261 | 0 |
| | Gamma (P.1) | 0 | 0 |
| | Omicron (B.1.1.529) | 11,657 | 429 |
| | Unknown [‡] | 49,786 | 0 |

*E* gene, envelope gene; VoC, variant of concern.

*For all variables except testing, data presented as first positive result per person.

[†]Does not include individuals who received ≥3 doses of vaccine.

[‡]Due to laboratory testing algorithms, only a selected portion of SARS-CoV-2-positive samples underwent characterization to identify the VoC lineage.

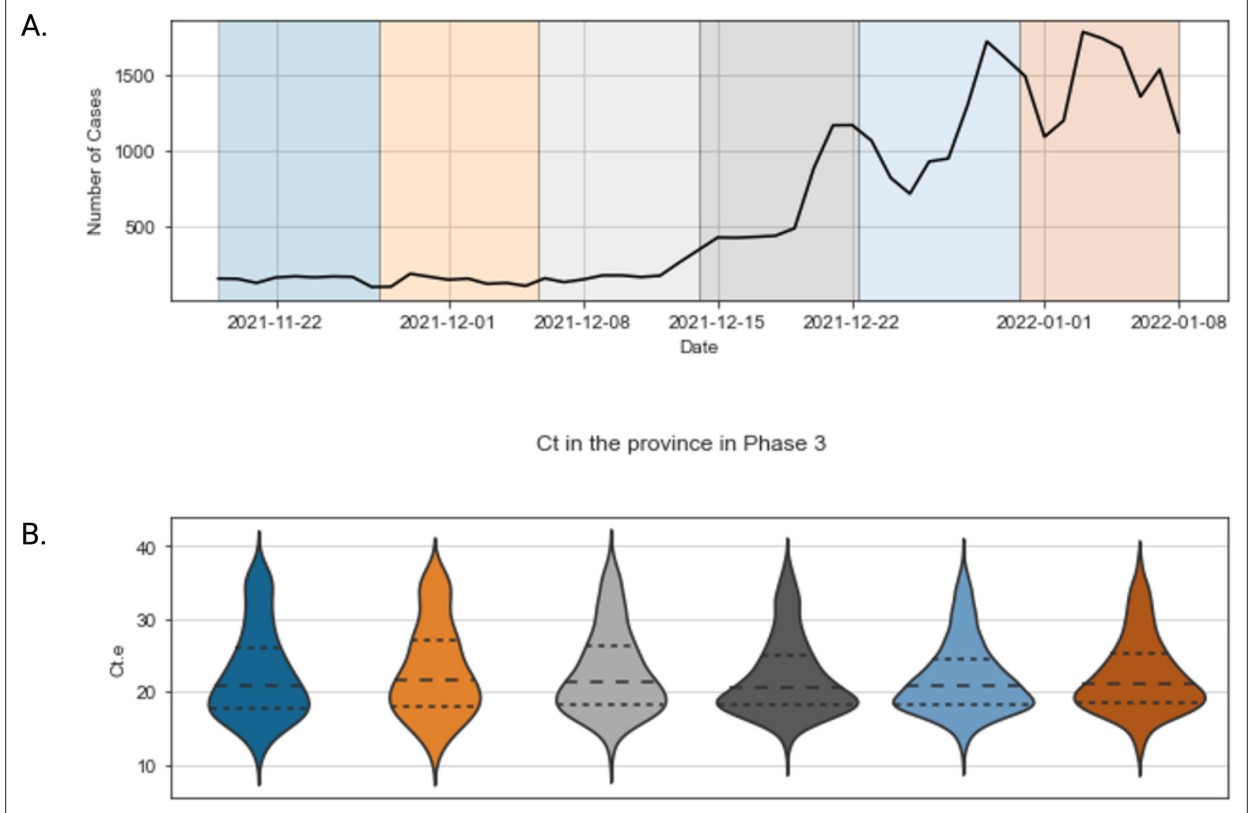

**Figure 2.** SARS-CoV-2 case incidence and E gene Ct distribution across study phases. Violin plots demonstrating the *E* gene cycle threshold value distribution in British Columbia across different time points of the study period (**A**). The median Ct values with associated interquartile range in dotted lines are presented. The absolute number of cases of confirmed SARS-CoV-2 infection is presented separately (**B**). Ct. e: Envelope (E) gene cycle threshold value; SARS-CoV-2: SARS-CoV-2: severe acute respiratory syndrome coronavirus type 2.

shows a decrease in Ct value on day 14, then showed an increase in Ct values from day 16 onward as the outbreak waned (*Figure 4C*).

## Simulated data: SEIR model applied to population-level data

The virosolver package was used to create simulated Ct data from an SEIR process on a sample population of 500,000 individuals. The default priors for the package were used to simulate an epidemic for 200 days with an observation period of 140 days to mimic real-world settings. A single cross-sectional SEIR model was fitted to the simulated Ct data using the MCMC method in the virosolver package. The predicted $R_t$ follows the true $R_t$ closely (*Figure 5*). The MSE of the SEIR model was 0.62% (95% confidence interval, 0.60–0.64%).

## Simulated data: Machine learning analysis

The fitted ML models were applied to out-of-sample data from the simulated Ct values, and the predicted $R_t$ of the five ML models were compared against the true simulated $R_t$ across sample sizes 100, 1000, and 10,000 (*Figure 6A*). With increasing sample size, the predicted $R_t$ with the associated 95% credible interval for the top performing model followed the true $R_t$ more closely (*Figure 6A*). Separately, the MSE was computed for each ML model comparing the predicted $R_t$ with the true $R_t$ (*Figure 6B*). Across all ML models, lower MSE (improved performance) was observed with increasing sample size (*Figure 6B*). The top-performing model at sample size 100 was LGBM with a median MSE distribution of 0.14 (0.03). The top-performing ML model for sample sizes 1000 and 10,000 was Random Forest, with a median MSE distribution of 0.05 (0.007) and 0.02 (0.003), respectively. The MSE for the Random Forest model decreased by 82% from sample size 100–10,000 demonstrating improved performance of the moments of the Ct distribution to predict $R_t$ on larger sample sizes. Each of the moments was examined for feature ranking importance through SHAP analysis. Across all ML

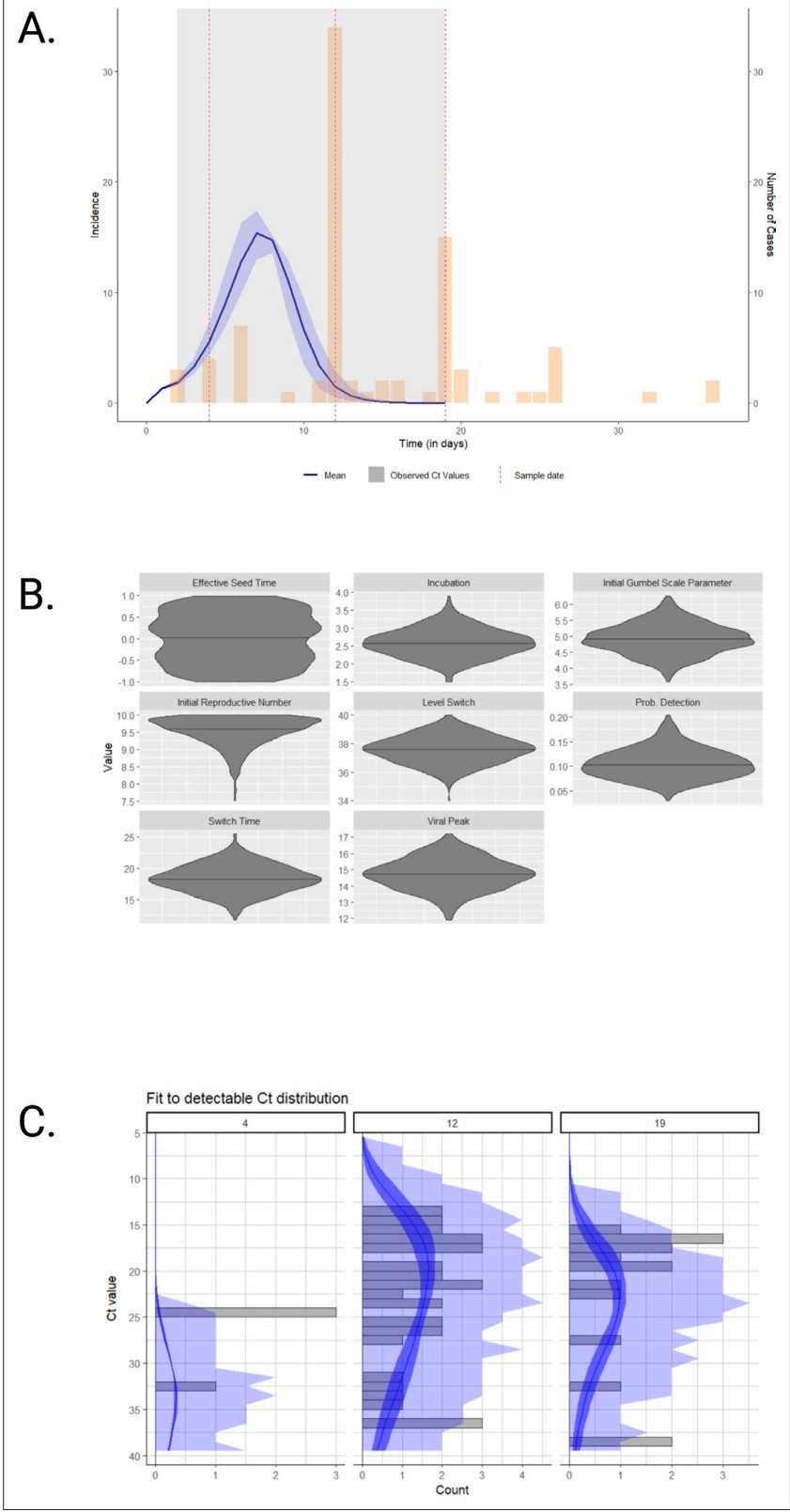

**Figure 3.** Long-term care facility outbreak investigation modeling findings. A multiple-cross-section SEIR model was fitted to the outbreak data (**A**) and showed a peak in incidence on the 12th day of the outbreak that preceded by 2 days the observed peak at the outbreak facility. The population included in this outbreak investigation was sampled at three predetermined time points (dashed red lines). The Monte Carlo chain model-predicted incidence

*Figure 3 continued on next page*

*Figure 3 continued*

curve is represented by a blue line and was overlaid with the reported number of confirmed SARS-CoV-2-positive cases in this outbreak setting in yellow bars. The blue ribbon represents the 95% credible interval. Violin plots of the viral kinetic parameters for the SEIR model are also presented in the outbreak case study (**B**). The MCMC approach searches over the viral kinetics described above and is based on prior values. Fit to detectable Ct distribution across time points of days 4, 12, and 19 is also presented in the outbreak study (**C**). These show the model fit (blue curve) overlaid with the frequency of Ct values (gray bars) and are a good indicator of the Ct distribution across the time points. The darker blue ribbon represents the 95% credible interval. The Ct values increase from outbreak days 12–19 as the epidemic declines. Ct: cycle threshold; SARS-CoV-2: severe acute respiratory syndrome coronavirus type 2; SEIR: susceptible-exposed-infected-recovered.

The online version of this article includes the following figure supplement(s) for figure 3:

**Figure supplement 1.** Case study epidemiological data (**A**) and epidemic curve (**B**) for the 156 infected individuals in the long-term care facility outbreak.

---

models and sample sizes, the variance of the Ct distribution was the top-ranking feature (*Figure 7—figure supplement 1*). Finally, the models presented relative advantages and disadvantages, which may impact feasibility for implementation and which are summarized separately (*Table 2*); however, this was limited by lack of ML analysis for the real-world data. Based on the results above, ML was found to be better suited for larger sample sizes and was flexible in design but presented greater computational complexity for analysis.

By comparing the ML (*Figure 7*) with the SEIR model (*Figure 5*) on the same simulated data, the SEIR showed better performance compared to all ML models. The SEIR model presented an MSE of 0.62% (95% CI, 0.60–0.64%) and the best performing ML model presented an MSE of 54% (95% CI, 39–83%) (*Table 2*).

## Discussion

In this study, we investigated the utility of two distinct modeling approaches based on cycle threshold values, epidemic transmission modeling, and machine learning, for incidence prediction and $R_t$ estimation, respectively. The SEIR model provided reasonable estimates for randomly sampled outbreak data and at a wider level on provincial data for the asymptomatic subgroup, and Random Forest performed with favorable accuracy on simulated data across the suite of five ML models. As testing needs overwhelmed laboratory capacity with increasing case burden and the emergence of variants of concern, molecular testing practice recommendations shifted to testing individuals who were symptomatic and/or with a minimal illness severity, resulting in sampling of a specific population. These changes in testing indications, foremost predicated on symptom-based testing, led to substantially more limited capacity to assess case counts for epidemic monitoring, generating a critical unmet need for other approaches to infer epidemic trends to support clinical and public health planning.

This study comprehensively investigated two modeling approaches, drawing on both previously published work and description of the novel application of machine learning modeling for SARS-CoV-2 transmission dynamics prediction. Our work highlights the need for these approaches to utilize data that are not conditioned on disease severity or other indicators, which impact the time between infection and specimen collection. For example, the long-term care facility dataset used for inference in the study required that specimens were collected randomly and not based on symptoms.

The rate of sampling is also important to consider with regards to accuracy of constructed incidence. Results from the long-term care facility indicate this modeling approach demonstrated a difference in incidence peak timing and amplitude across the horizons. This is likely explained due to the lag time between onset of the infectious period and reporting, given that site-wide facility testing was performed at set time periods rather than on a daily basis. We believe that these cases were produced through a combination of symptom testing and screening of the whole facility-wide, and thus the pattern of cases would not reflect the underlying incidence. However, this represents a pragmatic approach to real-world settings where these testing approaches would coexist. Thus, the model is estimating incidence rather than case number given the irregular testing performed. This supports the utility of this approach for other similar settings such as long-term care or assisted living or community-living facility outbreak investigations such as shelters, or within small hospital

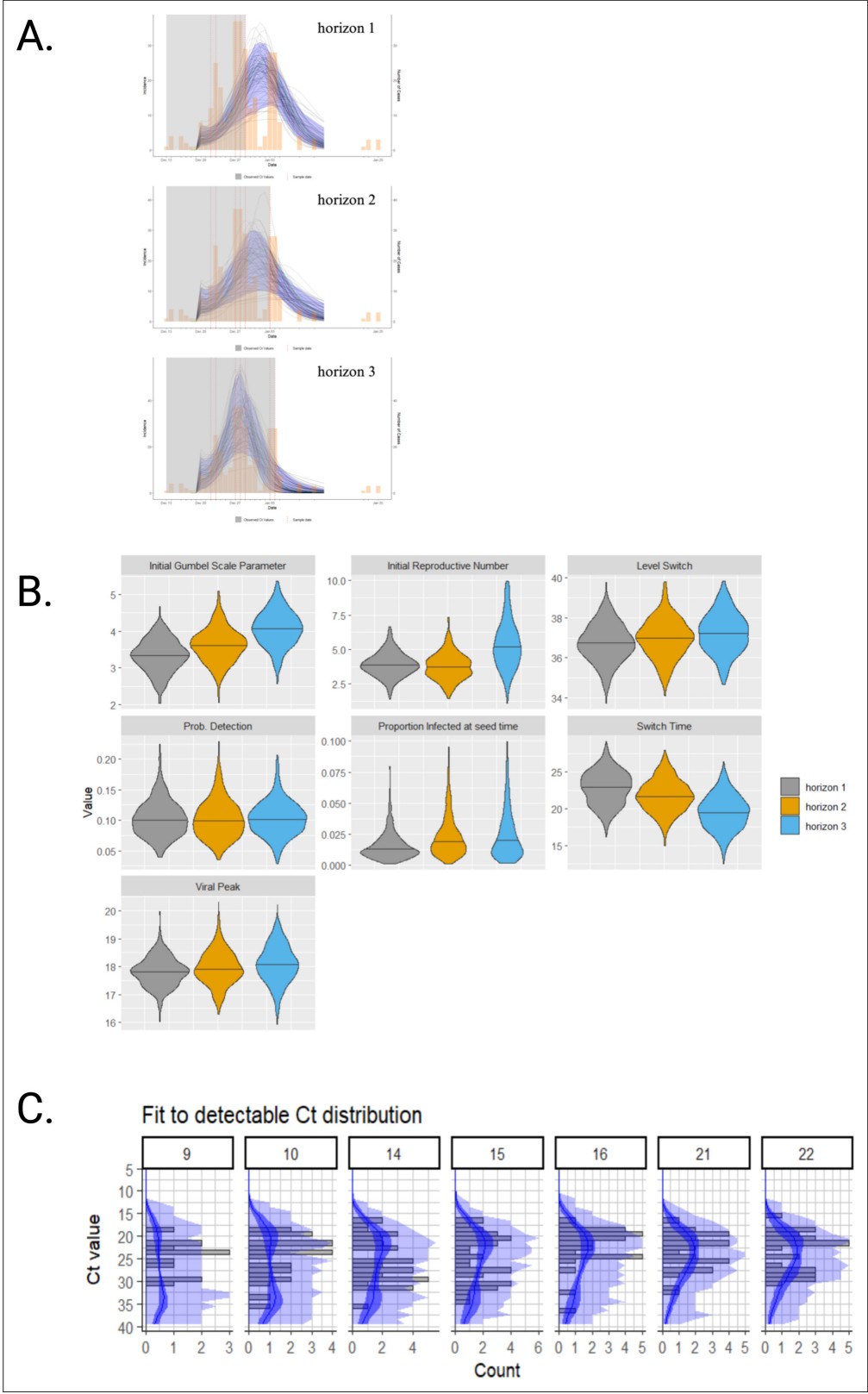

**Figure 4.** Overall population modeling findings. A multiple-cross-section SEIR model was fitted to the overall population-level data (**A**) and showed an incidence peak from December 27, 2021, to January 1, 2022, which overlapped with the observed peak of reported cases in the province. The Markov chain Monte Carlo model-predicted incidence curve is represented (black lines) and was overlaid with the reported number of confirmed

*Figure 4 continued on next page*

*Figure 4 continued*

SARS-CoV-2-positive (yellow bars) cases. Violin plots of the viral kinetic parameters for the SEIR model are presented (**B**). Three unique time horizons were chosen of sizes 5, 6, and 7. The MCMC approach searches over the viral kinetic parameters presented above and is based on prior values described separately (*Hay et al., 2021*). To align with the described Omicron viral kinetics, the incubation period was fixed and set at 3 days, and the infectious viral kinetic parameter was fixed. An upper bound of $I_0$ was set at 0.100. The initial reproductive number ($R_0$) increases across more horizons, which in turn shifts the SEIR peak earlier. The fit to detectable cycle threshold distribution is presented over the largest horizon (**C**). The largest frequency (gray bars) of model fit lowest Ct values (blue curve) occurs on days 14, 15, and 16, which represented the peak of the epidemic. The darker blue ribbon represents the 95% credible interval. Ct: cycle threshold; SARS-CoV-2: severe acute respiratory syndrome coronavirus type 2; SEIR: susceptible-exposed-infected-recovered.

systems. In contrast, the novel application of machine learning approaches described in this study showed improved fit with large datasets (such as >1000 COVID-19 positive cases), making this suitable for large population settings such as at the province, state, or large hospital network system level. Indeed, machine learning models can offer greater flexibility by incorporating different summary statistics and other data as features, fully harnessing the potential of larger datasets.

Importantly, the approaches described in this study demonstrated utility for timely prediction of SARS-CoV-2 transmission dynamics that could be harnessed to help inform future outbreak resource allocation and decision-making. For example, use of these models could be used to support decision-making across several settings, including hospitals, long-term care facilities, public health departments, and others, to help inform planning of resource allocation, vaccination efforts, and isolation practices. More specifically, this approach lays the groundwork for a sentinel surveillance monitoring strategy that could be automated and alert appropriate authorities at predetermined signals of predicted incidence changes and may be expanded to other infections for which testing is widespread and predictive tools are needed.

This study focused on a time period of Omicron (BA.1) predominance, and in the context of a sampled population with heterogeneous vaccination status, demonstrated accurate prediction of

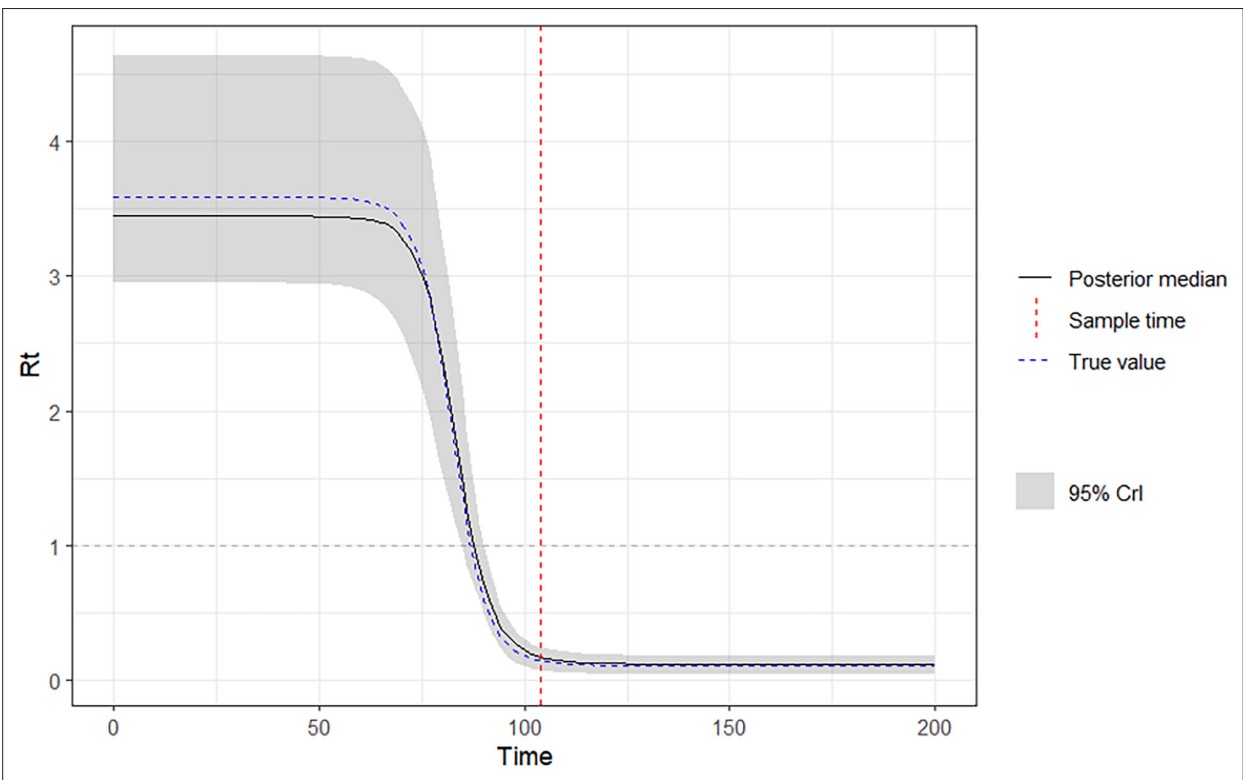

**Figure 5.** True $R_t$ (solid black) vs predicted $R_t$ (dotted blue) on a single-cross-sectional SEIR model using default viral kinetics. Sample time (t=104) (dotted red),.

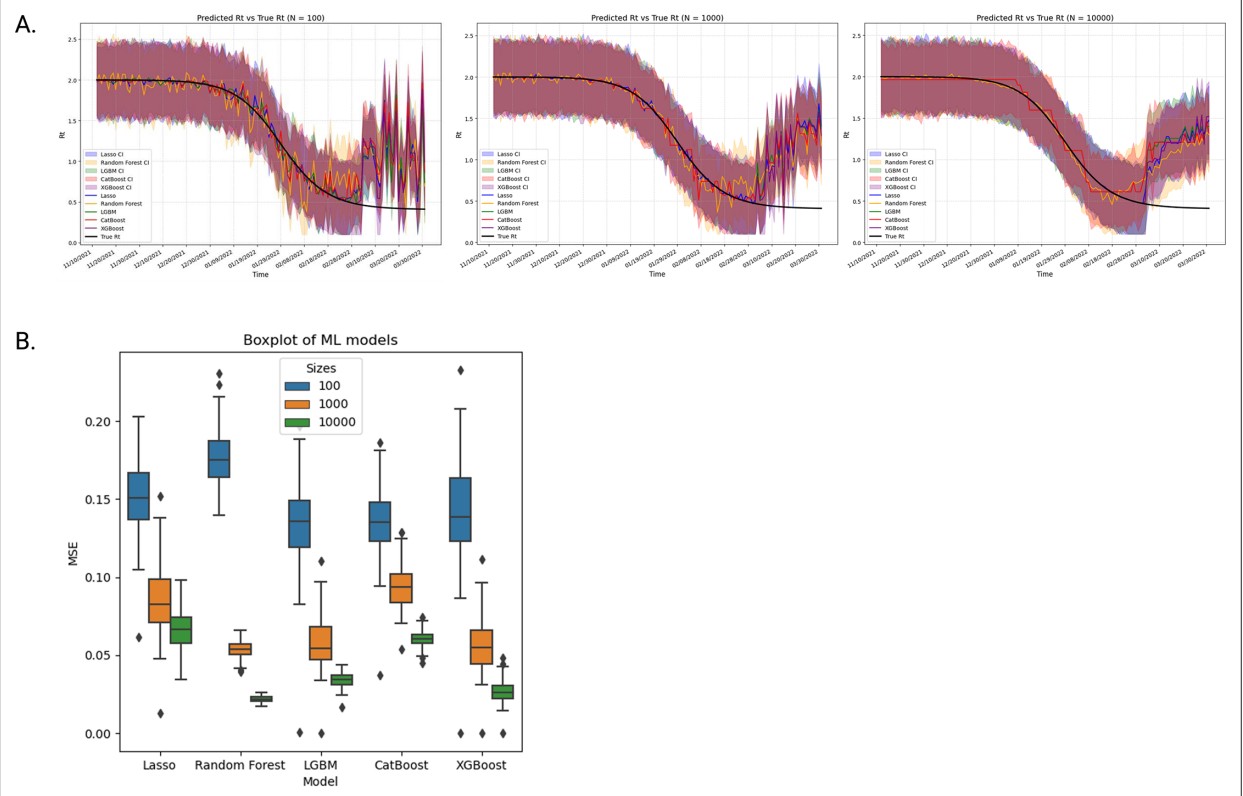

**Figure 6.** Effect of sample size on machine learning model prediction of the reproductive number. (**A**) Predicted $R_t$ vs True $R_t$ across all five machine learning (ML) models on three different sample sizes (n=100 [Panel I], 1,000 [Panel II] & 10,000 [Panel III]). The predicted $R_t$ across all models follows the true $R_t$ more closely with higher sample sizes. The 95% credible intervals are presented in colored ribbon. This is further corroborated by the (**B**) boxplot of the performance (MSE score) of all five models on three different sample sizes (n=100, 1000, and 10,000). Increasing sample sizes decreases the MSE, resulting in a more accurate predictive model. Random Forest is the best model at higher sample sizes.

incidence based on overall Ct distribution and viral kinetics without incorporating individual-level vaccination status. Further work is necessary to study the impact of vaccination status and other SARS-CoV-2 variants on the accuracy of incidence prediction.

One of the main strengths of this study is that it provides a comprehensive modeling toolkit that can be leveraged across settings and practical considerations for implementation. This approach could predict transmission dynamics in a way that could not be performed through case count analysis from biased sampling as was occurring in the province of BC. ML modeling is also advantageous as it can be performed in real time, rather than rely on monitoring of clinical indicators of severity such as hospitalization and intensive care unit admission that considerably lag behind true incidence rise. A limitation of previous studies is the use of a single or limited methodology for analysis that may perform well in a specific setting such as long-term care facilities, but lacked flexibility and predictive performance for generalizability to larger settings and in the context of changing testing practices (*Hay et al., 2021*). Additional strengths of this study include the independent assessment in a long-term care facility outbreak to validate the previously published model (*Hay et al., 2021*). This approach also lays the framework for expansion to use for other pathogens for which surveillance needs are critical, including other respiratory pathogens and possible integration with wastewater testing. Another consideration that we highlight in this study is the requirement of a sufficiently random sampling scheme to accurately estimate incidence from Ct value. Although there are limited covariates incorporated in the ML models, the first and second moments of Ct values suggested a predictive signal even in the absence of suitable viral kinetics.

Nonetheless, there are several limitations. Firstly, due to the insufficient number of individuals tested in the asymptomatic setting to perform ML analysis, we could not directly compare the performance of the two modeling strategies for real-world data. However, we focused on the comparison

between the two models for the simulated data and investigated key performance metrics for ML analysis on which future research may build. Further work will be required to fully characterize the relative advantages and disadvantages of each and to investigate the performance of ML models with more complicated Rt patterns. Second, the methodology used assumed random sampling that is challenging to confirm and is seldom achievable in real-world settings. Indeed, testing practices were modified following clinical and public health guidance of the province and may have led to bias in sampling. Restriction of the study population to the asymptomatic subgroup consisting of travelers and occupational health testing led to greater confidence in the employed sampling strategy tested and the validity of this assumption; nonetheless, there remains a need to develop a robust set of modeling approaches that can be leveraged across the broad variation of real-world sampling strategies. Third, even though the long-term care environment provides a more consistent testing environment, it tends to be a highly vaccinated population which may potentially introduce bias. In

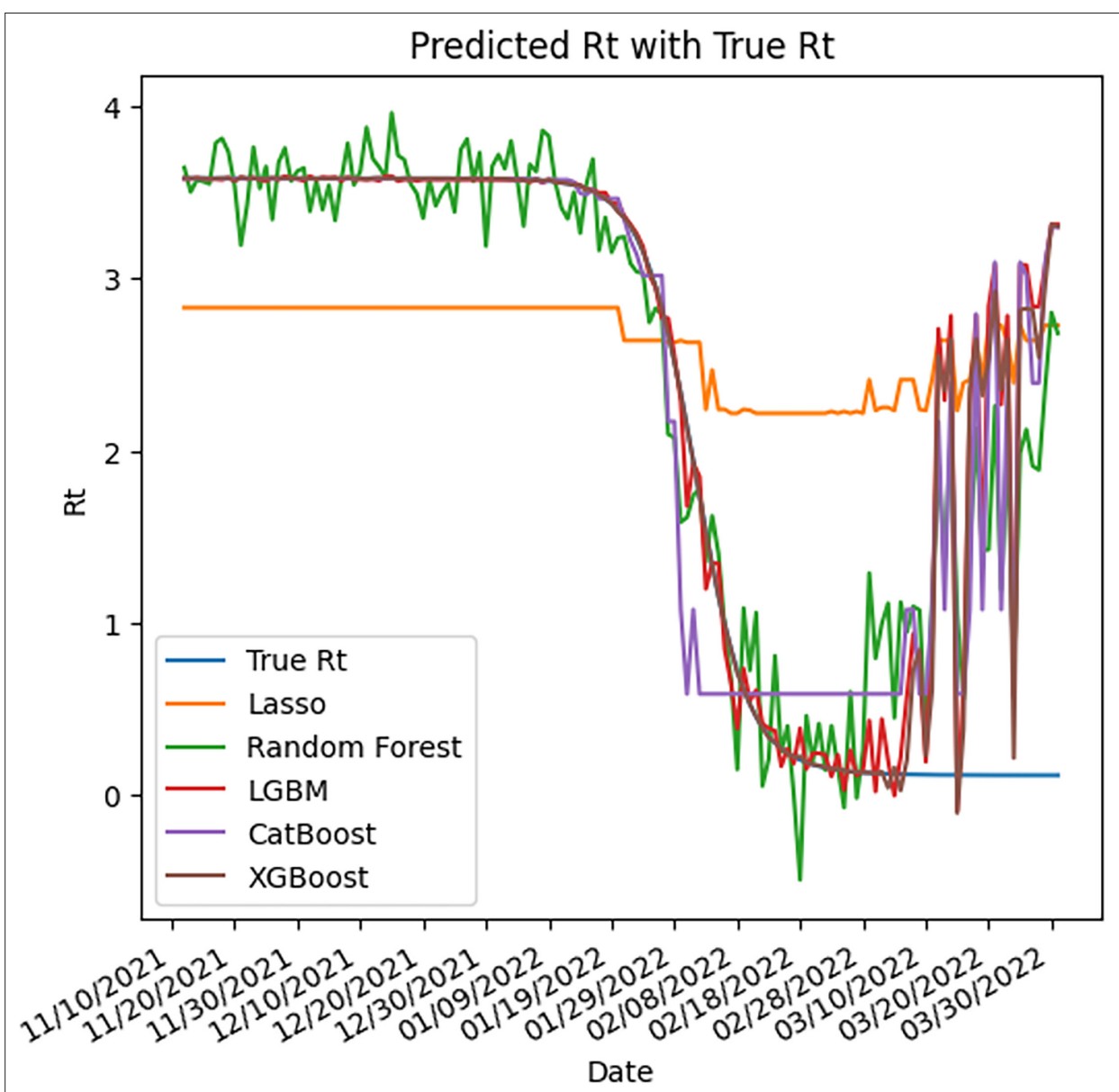

**Figure 7.** Predicted reproductive number ($R_t$) vs true $R_t$ (blue) across all models using the same simulation parameters from the SEIR model as presented in *Figure 5*.

The online version of this article includes the following figure supplement(s) for figure 7:

**Figure supplement 1.** Feature importance analysis by SHapley Additive exPlanations (SHAP) values.

**Table 2.** SEIR and ML model comparison across implementation considerations for SARS-CoV-2 incidence prediction.

| | Model | | SEIR | ML |
|---|---|---|---|---|
| Quantitative model comparison | Simulated data MSE (in %) | | 0.6 (95% CI, 0.60–0.64%) | Random Forest[†]: 54 (95% CI, 39–83%) |
| Qualitative model comparisons | Sampling type | | Random sampling | Random sampling |
| | Number of SARS-CoV-2-positive samples for which model best suited | | Small (»30) | Large (>1000) |
| | Sampling frequency | | Single/ multiple snapshots | Daily snapshots |
| | Flexibility | Modelling of transmission | Fixed in time | Time-independent |
| | | Ability to add in multiple predictors | Unable to incorporate | Able to incorporate, and flexible in their representation |
| | Scalability | | Single outbreak setting | Population level |
| | Computational complexity* | | Low | Low-moderate |
| | Predictive power requirements | | Good in single setting with well-mixed population and stable contact behaviour/ infection control | No requirements other than sufficient sample size for Ct summary statistics by snapshot |
| | Additional sampling requirements | | None | Ordered in time, restricted to fixed interval sampling |

Ct, cycle threshold; ML, machine learning; SARS-CoV-2, severe acute respiratory syndrome coronavirus type 2; SEIR, susceptible-exposed-infected-recovered.

*Relative computational complexity based on assumed sample size listed in Sacability row.

[†]Random Forest presented as was the top performing ML model.

addition to the above, complementary approaches that may be better suited for analysis of small populations, including stochastic modeling, should be investigated for future work. Fourth, this study did not leverage VoC or vaccination data, which are important potential confounders and for which the impact on viral load dynamics should be explored. Similarly, the models did not account for COVID-19-specific features including quarantine and social distancing (*Flaxman et al., 2020*), nor for wider applicability of the methods including long COVID and more recent VoCs, which may be relevant for future work in this field. Fifth, as the sample in our study was built from a patient population of unknown denominator, we could not realistically show the prediction interval; rather, we used this sample to create a general estimate of incidence for the entire population. Finally, this study aggregated Ct-level data across more than one assay from a single gene target, which may not adequately capture intra- and inter-assay variation or other gene target experience.

In summary, this study proposes a suite of modeling strategies, epidemic transmission modeling, and machine learning, based on population-level Ct values to accurately predict SARS-CoV-2 transmission dynamics. These modeling approaches may be used in real time to guide clinical and public health interventions. Such tools are needed to estimate incidence in a manner that is independent of the biases associated with testing guidance and to complement traditional surveillance based on case numbers or clinical indicators. Further work will be needed to expand validation of the models based on larger datasets and different settings with newly emerging variants and to assess real-time predictive power for direct clinical and public health impact.

## Acknowledgements

We thank the laboratory teams (virology, bacteriology, and molecular) at the BCCDC Public Health Laboratory for their contribution toward testing, on which this research is based. We also thank the data analytics team for supporting the data infrastructure and review that enabled this work. Finally, we thank the British Columbia Association of Medical Microbiologists for testing and sharing samples and data that enabled province-wide data collection, and public health partners throughout the province for their dedicated effort to outbreak management and infection control, and for sharing outbreak-level data that supported this research. This work was supported by funding by Genome

BC, Michael Smith Foundation for Health Research, and British Columbia Centre for Disease Control Foundation to CAH. This work was also funded by the Public Health Agency of Canada *COVID-19 Immunity Task Force COVID-19 Hot Spots Competition Grant (2021-HQ-000120)* to MGR.

## Additional information

### Competing interests

Miguel Imperial: is affiliated with LifeLabs. The author has no other competing interests to declare. The other authors declare that no competing interests exist.

### Funding

| Funder | Grant reference number | Author |
|---|---|---|
| BCCDC Foundation for Public Health | Public Health Rapid SARS-CoV-2 Vaccine Research Initiative in BC | Catherine A Hogan |
| Genome British Columbia | Public Health Rapid SARS-CoV-2 Vaccine Research Initiative in BC | Catherine A Hogan |
| Michael Smith Health Research BC | Public Health Rapid SARS-CoV-2 Vaccine Research Initiative in BC | Catherine A Hogan |
| Public Health Agency of Canada | 2021-HQ-000120 | Marc G Romney |

The funders had no role in study design, data collection and interpretation, or the decision to submit the work for publication.

### Author contributions

Afraz Arif Khan, Conceptualization, Data curation, Software, Formal analysis, Supervision, Validation, Investigation, Visualization, Methodology, Writing – original draft, Writing – review and editing; Hind Sbihi, Conceptualization, Resources, Data curation, Software, Formal analysis, Supervision, Investigation, Visualization, Methodology, Project administration, Writing – review and editing; Michael A Irvine, Software, Formal analysis, Supervision, Validation, Investigation, Visualization, Methodology, Writing – review and editing; Agatha N Jassem, Supervision, Investigation, Methodology, Writing – review and editing; Yayuk Joffres, Data curation, Formal analysis, Investigation, Visualization, Writing – review and editing; Braeden Klaver, Data curation, Investigation, Visualization, Methodology, Writing – review and editing; Naveed Janjua, Conceptualization, Supervision, Writing – review and editing; Aamir Bharmal, Formal analysis, Supervision, Investigation, Writing – review and editing; Carmen H Ng, Formal analysis, Investigation, Visualization, Methodology, Writing – review and editing; Chris D Fjell, Data curation, Software, Writing – review and editing; Miguel Imperial, Amanda Wilmer, John Galbraith, Marc G Romney, Supervision, Investigation, Writing – review and editing; Susan Roman, Marthe K Charles, Conceptualization, Project administration, Writing – review and editing; Bonnie Henry, Supervision, Writing – review and editing; Linda MN Hoang, Conceptualization, Supervision, Funding acquisition, Writing – review and editing; Mel Krajden, Conceptualization, Supervision, Investigation, Project administration, Writing – review and editing; Catherine A Hogan, Conceptualization, Data curation, Supervision, Funding acquisition, Investigation, Visualization, Methodology, Writing – original draft, Project administration, Writing – review and editing

### Author ORCIDs

Afraz Arif Khan ⓘ https://orcid.org/0000-0002-5545-2851
Miguel Imperial ⓘ https://orcid.org/0000-0002-4061-7940
Catherine A Hogan ⓘ https://orcid.org/0000-0003-1977-253X

### Decision letter and Author response

Decision letter https://doi.org/10.7554/eLife.95666.sa1
Author response https://doi.org/10.7554/eLife.95666.sa2

# Additional files

## Supplementary files

Supplementary file 1. Vaccination phase definitions used for the study.

Supplementary file 2. Epidemiological, clinical, and laboratory data of the earlier British Columbia SARS-CoV-2 pandemic phases.

Supplementary file 3. SARS-CoV-2 diagnostic testing strategy based on the envelope (*E*) gene target and test result interpretation criteria used.

Supplementary file 4. Control table of values and priors for the SEIR model.

Supplementary file 5. Hyperparameter selection.

MDAR checklist

## Data availability

All code and scripts used for data processing, analysis, and figure generation are publicly available on GitHub (copy archived at *Khan, 2026*). This study used linked COVID-19 surveillance and administrative data for the BC population. Data linkages and analyses conducted by the BC Centre for Disease Control were authorized under the Public Health Act for public health surveillance and risk assessment. Ethics approval was obtained from the University of British Columbia Research Ethics Board, which waived the requirement for participant consent because no identifiable data were used. The datasets include personal health information and are protected under the Freedom of Information and Protection of Privacy Act and, where applicable, the Public Health Act. As a result, the data cannot be publicly shared due to confidentiality requirements and the risk of re-identification, even after deidentification. Researchers may request access through the BC Centre for Disease Control or relevant data stewards within the Provincial Health Services Authority. Requests require a formal application and are subject to review by data governance bodies. Commercial use may require additional approval. For record-level data access, requests can be submitted to: datarequest@bccdc.ca.

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
