## [Editor Report]

This useful study presents an application of modelling approaches for estimating SARS-CoV-2 epidemiological dynamics from polymerase chain reaction (PCR) cycle threshold data. The authors provide solid evidence for their claim that their analytical pipeline has potential utility as a complement to analyses of traditional surveillance data. The results will be of interest to mathematical modellers working on communicable diseases with access to PCR data.

---

## [Decision Letter]

**Decision letter after peer review:**

[Editors’ note: the authors submitted for reconsideration following the decision after peer review. What follows is the decision letter after the first round of review.]

Thank you for submitting the paper "Prediction of SARS-CoV-2 transmission dynamics based on population-level cycle threshold values: A Machine learning and mechanistic modeling study" for consideration by *eLife*. Your article has been reviewed by 3 peer reviewers, and the evaluation has been overseen by a Reviewing Editor and a Senior Editor.

Comments to the Authors:

We are sorry to say that, after consultation with the reviewers, we have decided that this work will not be considered further for publication by *eLife*.

There was a consensus amongst the reviewers that substantially more analyses including simulation studies were needed to support the claims being made. While the reviewers were of the opinion that, with time, all the concerns could be addressed (and a detailed breakdown of the changes that need to be made is provided), it was felt that it would be likely to take longer to make these changes than *eLife* typically allows for a revision.

*Reviewer #1 (Recommendations for the authors):*

The authors of this study sought to apply a suite of models (five machine learning models, and an SEIR dynamical model) to infer epidemiological dynamics from Ct values. While this method has been previously established, that previous study was limited to situations in which Ct values were randomly sampled from the population. This study instead attempts to explore the robustness of these methods across biased sampling strategies and in the context of vaccination and changing SARS-CoV-2 variants.

Strengths of this study include its interesting dataset, which is generally well described. The paper also seeks to answer an important question about how robust methods of inferring epidemiological dynamics from Ct values are to sampling strategies. Another strength is the inclusion of multiple modelling approaches.

Unfortunately, the study suffers from a number of weaknesses. Many of its conclusions are only poorly, or not at all, supported by the results. For example, the paper purports to explore approaches that are robust across testing practises, but they do not seem to systematically compare across testing practices. In particular, the ML models and the SEIR model are never directly compared, or even fit to the same dataset. The ML models are only fit to simulated Ct values, not real data, and in this case, it appears that the sampling was always random; the only thing that varied was the sampling size. The SEIR model is fit to two different datasets, but predictions are not really compared between the two and are not quantitatively compared. The paper is also very confusingly written and is at times misleading or ambiguous. This is particularly the case for the description of the modelling strategy, when several critical details are insufficiently described (for instance, how the simulated data is generated, how the SEIR model is fit to data, or how the epidemiological dynamics are inferred from Ct values). These aspects make it very difficult to fully assess the paper.

The authors achieve part of their aims, namely to apply a range of modelling methodologies to the prediction of epidemiological dynamics using Ct values, although the description of aspects of this would benefit from substantial improvement. I do not feel that other aims of the paper, such as assessing these modelling approaches across sampling strategies, are achieved.

The paper seeks to address interesting questions about how to estimate epidemiological dynamics in biased sampling regimes, or when vaccination rates and variants are changing. At the moment, the paper does not convincingly answer these questions and is unlikely to have a large impact on the field. But it is possible that with a more systematic approach and a clearer and more complete description of what was done, the paper could make an important contribution.

Overall: To improve the paper, the most important thing is to improve the clarity of the writing. Several critical aspects of the methods are not described at all, and others are described confusingly or ambiguously. The authors might also consider comparing all models (ML and SEIR) on all datasets (simulated, population, and care home). This would at least give an indication of which models are performing well in different contexts; at the moment it is hard to tell. They could also consider trying simulated datasets with different sampling strategies, to enable a more systematic examination of these when the ground truth is known (I'm not sure whether this is feasible within your approach to simulating data, as that is not described). Other things you could consider include comparing these approaches to other model-fitting approaches, for instance fitting to the case data. In any case, I think the manuscript would really benefit from a clear description of its aims, its methods, and how those methods address those aims. Lastly, one simple thing that would really help is to increase the resolution and font size of most figures; I could not read several of them, which made it impossible to interpret them.

Line-by-line comments:

• line 57: I believe Ct values are inversely proportional to the log of the amount of virus

• lines 68-71: The sentence about the growth phase being associated with high viral load burden (and vice versa) needs a reference or some justification, as it is not completely obvious.

• 82: VoC acronym already introduced

• 91: I believe you only use one SEIR model

• Supp table 1: Please could you indicate the VoC distribution in this table to help understand the phases? Also, I think the asterisks don't correspond correctly to the text.

• The resolution of all figures needs to be improved. For instance, it is impossible to read parts of Figure 1A

• Figure 1A: In addition to the resolution, the font on aspects of this needs to be increased in size, as it would currently be almost illegible

• Figure 1B: It is not clear what the numbers represent here. In the box with "31,556 positive" please could you also state the total samples after exclusion. In the final box, it is not clear what the denominator for the percentage is, as the total is more than the number positive, but fewer than the total included samples.

• 129: It might be useful to know what proportion were done at different labs

• 135: Why did you only include the E gene target?

• Supp Figure 2: Please increase the font size. Also could you please indicate the period under study, I think this figure contains a lot of extraneous information, as you only used data from November 2021 onwards?

• 233: Please describe how the simulated data are generated

• General question: How are vaccination and variant data used in this study (or are they used)? Are they included in the data simulation process, or the SEIR model?

• Related: It seems like a lot of the data are not actually used in any of the models, for instance, data from before November 2021 or around vaccination and variants. While some discussion of this is necessary to understand the context, at the moment there may be too much time describing data that is not used, and it also may be misleading as it could lead the reader to believe that the models include this data. If this data is being used in the models, please can you describe how it is being used in the Models section of the methods.

• 246: Earlier on you stated you used two epidemic models, but now it seems like you are only using one. Please could you correct whichever of these is inaccurate?

• 246: Although this model is published, please could you give details on it here, or in the supplement, so that the reader can understand what was done without reading the other paper.

• 246: Related, please describe how the Ct values are used in the model.

• 248: I'm not sure the deterministic SEIR model is appropriate for a small population such as this, where stochastic dynamics (and importations) are likely to be important – did you consider trying a stochastic model?

• 249: How was the model fitted? Much more details are required here, please.

• 250: What modifications were made? Please be explicit

• 250: Why did you not fit the ML model to the provincial data due to sampling, but then did fit the SEIR model to it?

• 259: I'm not sure how this sentence is related to Figure 2.

• Figure 2: Panel B appears twice.

• 268: ML models. The first sentence of this paragraph, and the caption for Figure 3, imply that the MSE is of the predicted Ct values. But the second sentence talks about skill in predicting epidemic trends. Which one is it? (I'm assuming the latter, but the text is confusing)

• 269: It's not clear what this sentence means, as the reduction in MSE differs by model. I am not sure I understand the statement about higher moments having an increased ability to predict trends either. Please could you rephrase this sentence?

• Supp Figure 5: Many readers will not be familiar with this type of plot – please could you give additional details on how to interpret it, either in the main text or in the caption.

• 280: "The most precise results were observed with sampling from a total of five horizons." – what does this mean? I don't think you ever define the horizons.

• Figure 4: increase the font size. It is impossible for me to interpret this figure as stands due to the resolution and font size. Please can you use descriptive names for the variables (e.g., "initial proportion infected" (?) rather than I0), or at least define the variable names in the caption.

• SEIR section in Results: Please can you do a bit more to help the reader interpret this figure. I don't think you talk about panel II – perhaps add some text referring to this too.

• The paper insinuates it is comparing ML approaches with the dynamical model, but never fits them to the same data. It might be interesting to do so (either the surveillance data, the simulated data, or both). It's not clear to me why the limitations of the surveillance data mean that the ML models cannot be used, but the SEIR model still can (if anything, wouldn't it be the other way around, as the SEIR model is making stronger assumptions about how the data were generated?) In any case, there seems to be little harm in trying both and comparing their predictions, as this is ultimately a purely predictive exercise.

• 293: You stated in methods that the outbreak was used to validate the model, but here you state you are fitting to it. Please clarify.

• 303: "the ML model performed best on randomly-sampled province-level data" – best compared to what? Also which ML model are you referring to?

• 305: "Within epidemic transmission models, the SEIR model performed highest with randomly-sampled outbreak data" – highest compared to what? I was under the impression that you only used one epidemic model.

• 319: I'm not convinced that you "comprehensively investigated varying sampling types." It would be worth doing so, or if not, removing this statement.

• 321: How did you identify that "diagnostic testing indication, sampling type, and the individual population tested are critical factors", and what are they critical for?

• 323: How did you establish that "model selection must be tailored to the epidemiological circumstances of testing"? And how should it be tailored to the circumstances?

• 325: "SEIR modeling was only suitable for random sampling." – I don't believe you have established this. If I understand correctly, you fit this model to two datasets, one of which was randomly sampled (the care home dataset). Visually, the model appeared to fit the population (non-randomly sampled) dataset better.

• 327: "difference in peak timing" – difference from what? Do you mean across the horizons?

• 333-336: This sentence is misleadingly written. It suggests that the ML models work better with large datasets than the SEIR model – but you did not attempt to fit the SEIR and ML models to the same dataset. Instead I think you showed that the ML models show improved fit when a larger sample of the Ct values were used to train the model.

• 340: "all approaches described in this study could predict future trends within a one- to four-week timeframe" – I'm not sure you established this. For instance, I did not see this for any of the ML models, and it was not really quantified for the SEIR model.

• 375: I believe you only fitted one model to the care-home data, so this sentence is misleading (or perhaps just a typo). Moreover, it is not clear to me why this approach is classed as validation, while the others are not, as it seems like in all cases you fit the models independently to the relevant data.

• More discussion of limitations is needed.

*Reviewer #2 (Recommendations for the authors):*

This study investigates how machine learning methods using cycle threshold (Ct) values – the semi-quantitative measurement output from most RT-qPCR tests – from routine laboratory testing for SARS-COV-2 can accurately estimate SARS-CoV-2 epidemic trends. The authors obtained an impressive province-level dataset of SARS-CoV-2 RT-qPCR testing on nasopharyngeal swabs collected during the BA.1 Omicron wave in British Columbia, Canada, and linked this to a vaccination status database to stratify Ct values by date, vaccination status and variant of concern (VoC). However, it is not clear how the vaccination and VoC data are used in the machine learning analyses, as the authors only describe the predictive performance of the machine learning models applied to simulated data. The study also applies an existing epidemiological modelling approach to estimate the incidence curve from a well-studied outbreak in a long-term care facility using Ct values, finding that with multiple random samples, the underlying incidence curve for the outbreak could be estimated accurately. The epidemiological model is also applied to the province-level BC data to estimate a plausible infection incidence curve.

Overall, I do not think the authors have sufficiently achieved their aims, as the most robust results are largely an off-the-shelf application of an existing method to new datasets. There are also several conclusions that are not supported or contradicted by the results, often with no corresponding analyses presented. For example, the authors describe data on vaccination status and variants of concern in detail and mention that these data are included in the machine learning models (L371). However, the only results regarding the performance of the machine learning models use simulated data, and thus it is not clear how and where these additional real data are included. Another example is that the authors describe early on the focus on data from a restricted time period where most tests are expected to be in asymptomatic individuals (L105). However, on L231 the authors state that simulated data was used for the analysis due to testing guidelines tailored to symptomatic individuals. These and other inconsistencies weaken the claims of the paper.

Strengths:

The two main data sources for this study – the Provincial Health Laboratory Viewer and Reporter (PLOVER) and the Provincial Immunization Registry (PIR) dataset – have large sample sizes and if used as described would be fantastic datasets for testing these methods. The application of multiple machine learning and epidemiological modelling approaches to the same task of estimating the reproductive number and/or incidence using Ct values would be interesting if the accuracy of these various approaches were compared more robustly. I think this could be an important contribution. If the task of infectious disease surveillance is to accurately predict epidemic trends, then it is important to use state-of-the-art prediction methods like those used here. I also commend the authors for including the modelling code and simulated data on GitHub, which appears to be sufficiently documented.

Weaknesses:

The main weakness of this study is that many of the conclusions and described analyses are not presented in the manuscript as described above. It is often unclear which findings are related to real-world data and which are based purely on simulated data. For example, the reader is given the impression that the ML models are trained on simulated data and then tested on the real-world data to estimate the effective reproductive number for BA.1 Omicron in British Columbia during Nov/Dec 2021 (see L44, the Results section of the abstract). However, this analysis is never shown. The authors also claim to be addressing limitations of previous methods which do not account for changes in viral load kinetics due to vaccination and different variants, and non-random sampling strategies. However, none of the analyses this study presents account for these factors either. There are other studies on this topic that also consider non-random sampling, for example, Lin et al. Nature Communications 2021 https://doi.org/10.1038/s41467-022-28812-9.

Another weakness of the study is the lack of information with which to appraise the performance of the machine learning approaches for predicting the effective reproductive number. First, the only figures on predictive performance are the MSE results in Figure 3, which do not demonstrate how well the approach reconstructs the incidence curve. Second, as far as I can tell the ML approaches assume complete reporting of the Ct distribution statistics each day, which would not be feasible in reality.

The presented figures are hard to read, largely because the resolution is too low to read the smaller text (this can be corrected). Furthermore, the presented posterior estimates for the viral kinetics parameters in Figures 4 and 5 are impossible to interpret without further explanation/context unless one is already completely familiar with this model. The authors state that the values for these parameters were modified to fit the Omicron period, but what these modifications were and the rationale behind them are not explained.

I also find the inclusion of all the Ct distribution moments and the interpretation of their variable importance questionable. These metrics are highly correlated (e.g., mean and median), and thus I would expect there to be some more care in choosing which statistics to include. I also wonder if the importance of variance (as per the SHAP plots) may be because the methods are indirectly picking up sample size (i.e., number of positive tests) from variance – when there are very few positives then the empirical variance of the Ct distribution may be very low, but when there are many positives then the full spectrum of Cts may be represented.

Finally, while a semantic point, the study purports to investigate *prediction* of epidemic trends. The term prediction in infectious disease modelling is usually reserved for analyses seeking to extrapolate trajectories into the future. All the analyses presented here deal with nowcasting or hindcasting – estimating epidemic trends that have occurred up to the latest sampling date. Thus, I would not consider the methods here to be tackling a prediction task.

Impact on the field:

While the aims of this study are important to the field of infectious disease modelling and surveillance, the analyses would need substantial revision and extension to provide new insights beyond those already shown by previous work. The use of machine learning methods applied to Ct value distributions to predict infection incidence is an interesting topic, and understanding how time-varying vaccination coverage and circulation of variants with different viral kinetics impact these Ct distributions is interesting. However, this study is currently inadequate to address those questions.

I think this is a worthwhile study and the province-level BC dataset sounds fantastic. The availability of code is good, though the data are not provided (only a link to where they can be requested). I think the analyses and presentation would need to be substantially overhauled to support the claims made here. In addition to the broad points discussed above, some suggestions for improving the paper and specific areas which need clarifying are:

1. The ML results feel very incomplete and confusing. First, the mentioned analyses considering vaccination and VoC data should be presented and interpreted. Otherwise, I am not sure why those data are described and presented. Second, it needs to be much clearer where the ML methods are using simulated or real data. The idea of training the ML model on simulated data and then applying it to the real dataset is interesting, but I did not see any of those results. Third, I would suggest showing the epidemic curves or Rt estimates outputted from the ML models rather than just the overall MSE summary plots.

2. To claim that the study is investigating the prediction of epidemic trends rather than just reconstruction would require a formal assessment of prediction at future time horizons. On L340 you state that "all approaches described in this study could predict future trends within a one to four-week timeframe", but I see no results to support this claim.

3. It would be useful to see results on how well the ML models predicted (rather than just nowcasted/hindcasted) the real and simulated epidemics at different time horizons, sampling frequencies, and under different epidemic patterns rather than just simple SIR-like shapes. The results here show that the epidemiological model from virosolver performs quite well for the datasets used here. This will not be the case in most settings for the reasons you discuss in the introduction, and thus understanding where the mechanism-agnostic ML methods can outperform the epidemiological model would be useful.

4. The presentation of all the viral kinetics parameter posteriors is very odd (Figures 4 and 5), and I think would confuse most readers. I am very familiar with that work so understand what you are showing, but I think most readers would not. For example, the parameter names in Supplementary Table 6 are not interpretable without more context. Indeed, those parameters are largely "nuisance" parameters, and if the posterior estimates are biologically plausible and do not deviate too far from the priors, then they are not the main outputs of interest. It was also unclear to me what modifications to these parameters were made and why – were there some references on Omicron viral kinetics that were incorporated? The priors are also not shown here, only the upper and lower bounds. The columns "steps", "lower start" and "upper start" have no relevance to the reader and are specific to the virosolver code.

*Reviewer #3 (Recommendations for the authors):*

The authors study PCR (polymerase chain reaction) tests for COVID-19 in Canada, with investigation of the Ct (cycle threshold) on the tests. They use a variety of machine learning techniques to estimate the reproductive number (Rt) of COVID-19, based on received tests, and Ct. They validate their approach using "out of sample" data (i.e., holding out some of the data, and then predicting it.) This work is quite interesting, as Ct level data is under-studied. In addition, the problem is approached in a very pragmatic way in which many techniques are considered, and the results are well compared between methods. The methods considered are quite appropriate (for example, there is no need to have deep learning for such a square dataset.) Their results are strong.

As the authors mention, usually PCR test results are reported as positive/negative, based on a threshold. In this work, Ct is the "input" and Rt is the "output". This work is an insight into rich data about COVID-19. The work also investigates specific outbreaks within BC, Canada, and also considers vaccine status, and shows that their Rt estimations are good. This work examines sophisticated datasets, so we must value the insights that are gained from such close study of rich data.

In addition to validation on real data, this work also relies on simulations, and on the SEIR compartment model. This work could be improved by using compartment models more specific to COVID-19 (for example, with additional components for asymptomatic, or physical distancing), or more comparison with existing baselines. Further, estimation of Rt is not currently the most pressing concern with COVID-19 (can this work say anything about PACS, or MIS-C?)

- The reliance on simulated data is too strong. For example, in Figure 3, the machine learning methods are compared. It appears that all of the data going into this main comparison between machine learning methods is based on simulated data. Also, I struggle to see what the y-axis is on this figure (it is labelled MSE, but there are no units given). "First modeling approach: machine learning. The fitted ML models were applied to out-of-sample Ct data from the simulated Ct values (Figure 3). With increasing sample sizes, the MSE across each model reduced by 82% showing … " the reference to this main figure does not tell me what is being measured on the y-axis.

- Similar to above, while there is much reliance on simulated data, we do not see any details about how the data were simulated. "Ct data were generated to simulate a sufficiently large random sample of a population using the virosolver package". Some details about this method would be appropriate, or an indication of what parameter settings were used during the simulation, and why those parameters were used.

- Some parts of the paper rely on the SEIR model. This model is unrealistic for COVID-19, as quarantine, physical distancing, and asymptomatic cases are not considered. This work could be improved if a more sophisticated compartment model was used.

- Regarding the VOCs, in Table 1 we see 28,580 samples without a known lineage. Can this be further resolved, if not why?

- I do struggle a bit to know the relevance of the results: While any work on COVID-19 may have been in preparation for a long time, I wonder if Rt is currently an object of interest? I'd consider long-COVID, further VOCs, host genetics, or future pandemics as a more pressing research question with respect to COVID-19.

[Editors’ note: further revisions were suggested prior to acceptance, as described below.]

Thank you for resubmitting your work entitled "Prediction of SARS-CoV-2 transmission dynamics based on population-level cycle threshold values: An epidemic transmission and machine learning modelling study" for further consideration by *eLife*. Your revised article has been evaluated by John Schoggins (Senior Editor) and a Reviewing Editor.

The manuscript has been improved but there are some remaining issues that need to be addressed. In particular, new analysis is needed to address concerns raised in the first round of reviews and highlighted again in the review below. There are also still some claims that do not appear to be supported by the results as well as a number of areas where improvements in clarity and reporting of results are needed.

Reviewer #1 (Recommendations for the authors):

Thank you for the detailed response to my previous comments. I think the writing in this paper is now much improved and is less misleading and easier to follow than the previous version. Some of the claims are appropriately moderated in the discussion, and the methods are more complete and have a better structure.

However, I still have some misgivings about this paper as it is. It's a little disappointing the authors did not undertake any new analyses based on the lengthy comments provided by the reviewers in the previous round. In particular, I think the paper would really be much stronger if the authors were able to compare the modelling approaches on the same datasets – I don't quite understand why this is not possible. A greater variety of simulated data could also be used, to test the models' capabilities. Additionally, there are still parts of the manuscript which appear to compare the SEIR and ML models (e.g. L303). There are also parts of the manuscript (e.g. L85) which may lead the reader to believe that this paper tests these approaches under biased testing regimes (which would be an interesting analysis), but I believe it only ever employs the methods under a regime random sampling – unless I am missing something.

Some more specific line-by-line comments

• L80, 86-86: this still sounds like you are motivating the study in the same way as before – in terms of testing strategies and epidemiological settings. As the methods are the same as before, these things are still not established.

• L89: Please write out these acronyms when they are first introduced.

• The font size on a number of the figures is still very small – e.g. Figure 1A, 3A, 4A, …

• L189: I don't understand how it is possible that the model will fit real data but not simulated data. If the simulated data is of the same format as the real data, the process should be the same, surely? And as it's simulated data, it should be possible to generate it in the format you need?

• Relatedly, it's also still unclear to me why you can't fit the ML models to the province-level data – is it just a sample size issue? What happens when you try to fit the models?

• L201-207: just to be clear, the model is fit to the Ct value distributions, but not the case data, correct?

• L207: For the omicron adaptations, please could you give a reference for why you made these choices. It might also be useful to know what these parameters where in the original model.

• Figure 2A: Recommend you present the lower and upper quartiles, rather than just the width of the IQR.

• L264 – Rt is not measured in days

• Figure 3A: It might be easier to see what is going on if you show mean and 95% credible interval, but not the sampled trajectories which obscure the 95% CrI

• L260 and L274: what does it mean to say the incidence is within the 95% CrI? It looks like sometimes it is, but often it isn't. Relatedly, in the captions for Figures3 and 4, you do not describe what the blue ribbon represents (I am assuming this is the 95% CrI)

• In Figure 3A and 4A, could you also show the prediction interval (i.e. accounting for case reporting?) This might help to understand whether the estimated incidence matches the true incidence. Perhaps this is not possible with this model, in which case ignore me.

Relatedly, you could then quantify the fit by reporting the coverage of the 95% prediction interval (proportion of incidence values which fall within the 95% range)

• L303: Again, you don't compare the models (SEIR vs ML), so I just don't think you can make claims like this.

• L354: I may have missed it, but I don't believe you compared the Ct distribution of omicron to other variants.

• In several places in the results and discussion, I think you are overstating the success of these approaches. The SEIR appears to do quite poorly in the LTCF, and not outstandingly in the population-level data. The machine learning models similarly do okay, but not amazingly, even though the simulated context is likely to be much simpler than what would be encountered in real life, as it is based on a perfect epidemic curve.

• Relatedly, it would be interesting to try and fit the machine learning models to more complicated Rt patterns.

• Could you add confidence intervals to the Rt estimates from the ML model, e.g. by bootstrapping or some other approach?

[Editors’ note: further revisions were suggested prior to acceptance, as described below.]

Thank you for resubmitting your work entitled "Prediction of SARS-CoV-2 transmission dynamics based on population-level cycle threshold values: An epidemic transmission and machine learning modeling study" for further consideration by *eLife*. Your revised article has been evaluated by John Schoggins (Senior Editor) and a Reviewing Editor.

The manuscript has been improved but there is one remaining issue from Reviewer #1 that needs to be addressed.

Reviewer #1 (Recommendations for the authors):

I thank the reviewers for carefully addressing all of my comments. I have just one remaining query, after which I am happy with this manuscript:

One of my main comments on the previous round pertained to comparing the SEIR and ML models on simulated data. The authors state in their response that they do this in Figure 3; however, I can't see how figure 3 does this--it seems instead to be just the SEIR results fitted to the real world data? Or probably I am missing something! I think I need to compare Figures5 and 6? Please could the authors clarify where this comparison is shown.

---

## [Author Response]

[Editors’ note: the authors resubmitted a revised version of the paper for consideration. What follows is the authors’ response to the first round of review.]

Reviewer #1 (Recommendations for the authors):The authors of this study sought to apply a suite of models (five machine learning models, and an SEIR dynamical model) to infer epidemiological dynamics from Ct values. While this method has been previously established, that previous study was limited to situations in which Ct values were randomly sampled from the population. This study instead attempts to explore the robustness of these methods across biased sampling strategies and in the context of vaccination and changing SARS-CoV-2 variants.Strengths of this study include its interesting dataset, which is generally well described. The paper also seeks to answer an important question about how robust methods of inferring epidemiological dynamics from Ct values are to sampling strategies. Another strength is the inclusion of multiple modelling approaches.Unfortunately, the study suffers from a number of weaknesses. Many of its conclusions are only poorly, or not at all, supported by the results. For example, the paper purports to explore approaches that are robust across testing practises, but they do not seem to systematically compare across testing practices. In particular, the ML models and the SEIR model are never directly compared, or even fit to the same dataset. The ML models are only fit to simulated Ct values, not real data, and in this case, it appears that the sampling was always random; the only thing that varied was the sampling size. The SEIR model is fit to two different datasets, but predictions are not really compared between the two and are not quantitatively compared. The paper is also very confusingly written and is at times misleading or ambiguous. This is particularly the case for the description of the modelling strategy, when several critical details are insufficiently described (for instance, how the simulated data is generated, how the SEIR model is fit to data, or how the epidemiological dynamics are inferred from Ct values). These aspects make it very difficult to fully assess the paper.The authors achieve part of their aims, namely to apply a range of modelling methodologies to the prediction of epidemiological dynamics using Ct values, although the description of aspects of this would benefit from substantial improvement. I do not feel that other aims of the paper, such as assessing these modelling approaches across sampling strategies, are achieved.The paper seeks to address interesting questions about how to estimate epidemiological dynamics in biased sampling regimes, or when vaccination rates and variants are changing. At the moment, the paper does not convincingly answer these questions and is unlikely to have a large impact on the field. But it is possible that with a more systematic approach and a clearer and more complete description of what was done, the paper could make an important contribution.Overall: To improve the paper, the most important thing is to improve the clarity of the writing. Several critical aspects of the methods are not described at all, and others are described confusingly or ambiguously. The authors might also consider comparing all models (ML and SEIR) on all datasets (simulated, population, and care home). This would at least give an indication of which models are performing well in different contexts; at the moment it is hard to tell. They could also consider trying simulated datasets with different sampling strategies, to enable a more systematic examination of these when the ground truth is known (I'm not sure whether this is feasible within your approach to simulating data, as that is not described). Other things you could consider include comparing these approaches to other model-fitting approaches, for instance fitting to the case data. In any case, I think the manuscript would really benefit from a clear description of its aims, its methods, and how those methods address those aims. Lastly, one simple thing that would really help is to increase the resolution and font size of most figures; I could not read several of them, which made it impossible to interpret them.

We thank the reviewer for this important feedback, and have substantially revised the manuscript to be more systematic and clearer, and have revised the figures as suggested.

Line-by-line comments:• line 57: I believe Ct values are inversely proportional to the log of the amount of virus

We have edited this sentence as suggested:

‘SARS-CoV-2 viral burden can be quantitated by the use of polymerase chain reaction (PCR) cycle threshold (Ct) values, which are inversely proportional to the log amount of target viral sequence present in the patient sample.’ (Lines 63-65)

• lines 68-71: The sentence about the growth phase being associated with high viral load burden (and vice versa) needs a reference or some justification, as it is not completely obvious.

We have added the Hay *et al.* reference at the end of this sentence (Line 77).

• 82: VoC acronym already introduced

This sentence no longer appears in the Introduction, and VoC is now only defined once (Line 96).

• 91: I believe you only use one SEIR model

This has now been corrected (Line 88).

• Supp table 1: Please could you indicate the VoC distribution in this table to help understand the phases? Also, I think the asterisks don't correspond correctly to the text.

The VoC information has been added in a new column, and the asterisks have been corrected.

• The resolution of all figures needs to be improved. For instance, it is impossible to read parts of Figure 1A

This has been corrected in the revised manuscript.

• Figure 1A: In addition to the resolution, the font on aspects of this needs to be increased in size, as it would currently be almost illegible

The font size has been increased across this figure.

• Figure 1B: It is not clear what the numbers represent here. In the box with "31,556 positive" please could you also state the total samples after exclusion. In the final box, it is not clear what the denominator for the percentage is, as the total is more than the number positive, but fewer than the total included samples.

We agree with the reviewer that this required major editing, and have revised this figure accordingly. Of note, all numbers have been edited to reflect the most recent iteration of analysis.

• 129: It might be useful to know what proportion were done at different labs

We have now included that testing was performed at two main sites (Line 117). Of note, Supplemental Table 2 was streamlined to include only laboratories that contributed real-world data for analysis.

‘Testing was performed at two main sites, and included the BCCDC PHL laboratory-developed test (LDT) (22) and the Panther Fusion SARS-CoV-2 assay (Hologic, Malborough, MA) (Supplemental Table 2).’ (Lines 117-119)

• 135: Why did you only include the E gene target?

We have now provided justification for this in the revised manuscript:

‘The *E* gene was selected as it was the most commonly tested target across the participating laboratory sites.’ (Lines 116-117)

• Supp Figure 2: Please increase the font size. Also could you please indicate the period under study, I think this figure contains a lot of extraneous information, as you only used data from November 2021 onwards?

We have now increased the font size for easier reading and outlined the study period by dotted vertical lines on the Figure. The study dates are also listed in the legend for ease of reference. Of note, we modified the order in the manuscript of the previous Supplemental Figure 1 (now Supplemental Figure 4), such that the current Figure is now referred to as Supplemental Figure 1.

• 233: Please describe how the simulated data are generated

We have now added this information in the revised manuscript:

‘To simulate infection times and Ct values, a separate deterministic SEIR model adapted from the virosolver package was used for the machine learning approach. Full detail on the data simulation is included in the Supplemental Data, and all code used in the current study to enable reproduction is provided separately (https://github.com/Afraz496/Vital-E-paper). The simulation sample period was set at 140 calendar days to encompass a typical single SARS-CoV-2 wave. Ct values were generated to simulate a sufficiently large random sample of a population, and was applied on sample sizes of 100, 1,000 and 10,000 on a simulated population of 50,000 individuals.’ (Lines 220-227)

• General question: How are vaccination and variant data used in this study (or are they used)? Are they included in the data simulation process, or the SEIR model?

We thank the reviewer for pointing this out, and agree that these are important to consider. However, given that the focus of the current study was on validating the Hay *et al.* model in a different setting, and as these two variables could not be incorporated in the data simulation process, VoC and vaccination data were only included in the descriptive analysis and not incorporated in the modelling. The ML pipeline is set up for incorporating these variables; however, the Hay *et al.* model will require extensive additional adaptation. This will require further work to investigate comprehensively. For enhanced clarity and based on this feedback, we have now removed the emphasis on VoC and vaccination data in the Introduction and Discussion in the revised manuscript, and more clearly acknowledge this as a limitation of the current study:

‘Fourthly, this study did not leverage VoC or vaccination data, which are important potential confounders and for which the impact on viral load dynamics should be explored.’ (Lines 395-397)

• Related: It seems like a lot of the data are not actually used in any of the models, for instance, data from before November 2021 or around vaccination and variants. While some discussion of this is necessary to understand the context, at the moment there may be too much time describing data that is not used, and it also may be misleading as it could lead the reader to believe that the models include this data. If this data is being used in the models, please can you describe how it is being used in the Models section of the methods.

We have streamlined relevant sections in the revised manuscript, including below:

‘COVID-19 testing practices changed over time in BC. From January 2021 onward, testing was prioritized for individuals at increased risk of severe disease or who worked in high-risk settings (20, 21). In addition, starting in December 2021, testing of asymptomatic and mildly symptomatic individuals was initiated with the organized roll-out of rapid antigen tests. During the time course of the study, COVID-19 molecular testing was performed at both the reference public health laboratory (BCCDC PHL) and at first-line laboratories across the province.’ (Lines 107-112)

In addition, as suggested above, we have also removed emphasis on VoC and vaccination in the Introduction.

• 246: Earlier on you stated you used two epidemic models, but now it seems like you are only using one. Please could you correct whichever of these is inaccurate?

This has now been harmonized to a single model (Lines 177-179).

• 246: Although this model is published, please could you give details on it here, or in the supplement, so that the reader can understand what was done without reading the other paper.

This has now been added:

‘In brief, this previously-published model uses population-level viral load distributions calibrated to known features of SARS-CoV-2 viral load kinetics to estimate the epidemic trajectory from single or multiple cross-sections of positive samples, and was initially validated on data from an outbreak in long-term care facilities in Massachusetts.’ (Lines 179-182)

• 246: Related, please describe how the Ct values are used in the model.

We have now added this information:

Referring to SEIR:

‘Using this approach, discrete Ct values are incorporated in the compartmental model over a series of time horizons. Horizons refer to time points across the sample period which draw on the Ct values to search across the Markov chain Monte Carlo (MCMC) chains to predict the incidence of the sample period.’ (Lines 182-185)

Referring to ML:

‘Using this approach, daily Ct value data are aggregated and incorporated as moments including mean, median, variance, skewness and kurtosis, rather than incorporated as discrete Ct values.’ (Lines 227-229)

• 248: I'm not sure the deterministic SEIR model is appropriate for a small population such as this, where stochastic dynamics (and importations) are likely to be important – did you consider trying a stochastic model?

We thank the reviewer for this comment. As we aimed to validate the previously-published method from Hay *et al.* in a separate but similar setting of facility outbreak, we applied the same methodology using a deterministic SEIR model for our study. We agree that a stochastic model that does not carry the implicit assumption of infinite population size as a SEIR model does, and would be a worthwhile extension. However, given known challenges with Bayesian inference on SEIR models, this was considered out of scope of the current study, and may be pursued in future work dedicated to this approach. We have now incorporated this in areas for future work in the revised manuscript:

‘In addition to above, complementary approaches that may be better suited for analysis of small populations including stochastic modelling should be investigated for future work.’ (Lines 393-395)

• 249: How was the model fitted? Much more details are required here, please.

We have now added this in the revised manuscript.

‘Thus, the current study SEIR model was subsequently fitted to province-level data from asymptomatic individuals using a MCMC framework. This uses a modified Metropolis-Hastings algorithm that incorporates discrete Ct values to generate univariate uniform proposals. Here, horizons were set to sizes 5, 6, and 7 based on separability (significant distance between time points) and availability of data. The SEIR model performance was considered accurate if the true incidence fell within the predicted incidence of the 95% credible interval of the MCMC chains.’ (Lines 201-207)

• 250: What modifications were made? Please be explicit

We have now described these modifications in detail as recommended:

‘Modifications to the viral kinetics for the SEIR model were applied to the provincial data to account for the specific nature of the Omicron (BA.1) variant (24, 25). The initial time (t0) was fixed to 1 day, the incubation time was fixed to 3 days, and the infectious period was fixed to the default value of 4 days. Fixed here implies that the viral kinetics were made static rather than dynamic by searching for the parameters via the MCMC framework. In addition, the model searched for I0, and the upper bound was set to 0.1 based on estimated provincial incidence during the timeline of the study.’ (Lines 207-213)

• 250: Why did you not fit the ML model to the provincial data due to sampling, but then did fit the SEIR model to it?

We agree with the reviewer that this would have been ideal. However, due to the data limitations described in the revised manuscript section below, we could not include this analysis.

‘A separate real-world data analysis by ML was planned for the study; however, due to the insufficient number of randomly-tested samples with which to conduct the study with real-world data, this could not be performed and only simulated data were used.’ (Lines 218-220)

• 259: I'm not sure how this sentence is related to Figure 2.

We have now removed emphasis on specimen type distribution in the results, and this sentence no longer appears in the revised manuscript.

• Figure 2: Panel B appears twice.

This has now been corrected.

• 268: ML models. The first sentence of this paragraph, and the caption for Figure 3, imply that the MSE is of the predicted Ct values. But the second sentence talks about skill in predicting epidemic trends. Which one is it? (I'm assuming the latter, but the text is confusing)

We agree with the reviewer that this was confusing, and have now extensively improved this section in the revised manuscript:

‘The fitted ML models were applied to out-of-sample data from the simulated Ct values, and the predicted Rt of the five ML models were compared against the true simulated Rt across sample sizes 100, 1,000 and 10,000 (Figure 5A). With increasing sample size, the predicted Rt for the top performing model followed the true Rt more accurately (Figure 5A). Separately, the MSE was computed for each ML model comparing the predicted Rt with the true Rt (Figure 5B). Across all ML models, lower MSE (improved performance) was observed with increasing sample size (Figure 5B). The top performing model at sample size 100 was LGBM with a median MSE distribution of 0.14 (0.03). The top performing ML model for sample sizes 1,000 and 10,000 was Random Forest, with a median MSE distribution of 0.05 (0.007) and 0.02 (0.003), respectively. The MSE for the Random Forest model decreased by 82% from sample size 100 to 10,000 demonstrating improved performance of the moments of the Ct distribution to predict Rt on larger sample sizes. Each of the moments was examined for feature ranking importance through SHaP analysis. Across all ML models and sample sizes, the variance of the Ct distribution was the top-ranking feature (Supplemental Figure 5).’ (Lines 289-302)

• 269: It's not clear what this sentence means, as the reduction in MSE differs by model. I am not sure I understand the statement about higher moments having an increased ability to predict trends either. Please could you rephrase this sentence?

We thank the reviewer for this feedback, and have rewritten this section as recommended, and as described immediately above.

• Supp Figure 5: Many readers will not be familiar with this type of plot – please could you give additional details on how to interpret it, either in the main text or in the caption.

The Supplemental Figure 5 has now been revised to include additional information to help with interpretation as suggested:

‘The colors indicate the association between machine learning outputs and simulated cycle threshold (Ct) data. Features that have a higher impact on the prediction of R_t_ are presented in pink, and features that have a lower impact on the prediction of R_t_ are in blue. Results are presented stratified by three different population sizes: 100, 1,000 and 10,000 with each column in descending order of performance. Of the five features explored, the top ranking feature across all models was the variance of the Ct data.’ (Supplemental Figure 5)

• 280: "The most precise results were observed with sampling from a total of five horizons." – what does this mean? I don't think you ever define the horizons.

We have now revised this section extensively for enhanced clarity in Methods:

‘Using this approach, discrete Ct values are incorporated in the compartmental model over a series of time horizons. Horizons refer to time points across the sample period which draw on the Ct values to search across the Markov chain Monte Carlo (MCMC) chains to predict the incidence of the sample period.’ (Lines 182-185)

and

‘Here, horizons were set to sizes 5, 6, and 7 based on separability (significant distance between time points) and availability of data.’ (Lines 204-205)

• Figure 4: increase the font size. It is impossible for me to interpret this figure as stands due to the resolution and font size. Please can you use descriptive names for the variables (e.g., "initial proportion infected" (?) rather than I0), or at least define the variable names in the caption.

Figure 4 has been revised as suggested, including with full variable name description.

• SEIR section in Results: Please can you do a bit more to help the reader interpret this figure. I don't think you talk about panel II – perhaps add some text referring to this too.

We have extensively edited the Results section for enhanced clarity, and now specifically reference to Figures 3B and 4B regarding panel II.

• The paper insinuates it is comparing ML approaches with the dynamical model, but never fits them to the same data. It might be interesting to do so (either the surveillance data, the simulated data, or both). It's not clear to me why the limitations of the surveillance data mean that the ML models cannot be used, but the SEIR model still can (if anything, wouldn't it be the other way around, as the SEIR model is making stronger assumptions about how the data were generated?) In any case, there seems to be little harm in trying both and comparing their predictions, as this is ultimately a purely predictive exercise.

We thank the reviewer for this insightful feedback, and completely agree that in a perfect case scenario we would have been able to present head-to-head data between the modeling strategies. However, despite our best efforts this could not be achieved for the current paper due to limitations that are now clearly described the Methods and Discussion:

Regarding SEIR limitation:

‘Application of the SEIR model in the current study was first attempted on simulated Ct data. However, the current version of the virosolver package does not support SEIR modeling on simulated data (https://github.com/jameshay218/virosolver_paper/issues/2); as such, this analysis could not be pursued. Rather, the SEIR model was validated on data from a long-term care facility outbreak that occurred in BC where point prevalence testing was performed at infrequent intervals as described above.’ (Lines 187-192)

Regarding ML limitation:

‘The second modeling approach was based on a collection of ML approaches for prediction of the reproductive number on simulated data, including Lasso (26), Random Forest (RF), Light Gradient Boosting Modeling (LGBM), eXtreme Gradient Boosted Modeling (XGBM) and CatBoost. A separate real-world data analysis by ML was planned for the study; however, due to the insufficient number of randomly-tested samples with which to conduct the study with real-world data, this could not be performed and only simulated data were used.’ (Lines 215-220)

Regarding Discussion:

‘Firstly, due to limitations in the models used and insufficient numbers of individuals tested in the asymptomatic setting, we could not directly compare the performance of the two modelling strategies. However, we comprehensively investigated each approach and emphasized key performance metrics for each. Further work will be required to generate a head-to-head comparison, and fully characterize the relative advantages and disadvantages of each.’ (Lines 379-384)

• 293: You stated in methods that the outbreak was used to validate the model, but here you state you are fitting to it. Please clarify.

We have revised the wording of this section extensively, as follows in Results:

‘A single horizon with three time points was used for constructing the multiple cross-section SEIR model on the outbreak data. This model showed a peak in incidence on the 12th day of the outbreak which preceded by two days the observed peak at the outbreak facility (Figure 3A). The SEIR model demonstrated accurate prediction with the real incidence falling within the 95% credible interval of the predicted MCMC chains. The model also accurately predicted the decline in cases by the 20th day of the outbreak (Figure 3A).’ (Lines 256-262)

• 303: "the ML model performed best on randomly-sampled province-level data" – best compared to what? Also which ML model are you referring to?

We have revised this sentence as follows:

‘The SEIR model performed with high accuracy for randomly-sampled outbreak data and at a wider level on provincial data for the asymptomatic subgroup, and Random Forest performed with high accuracy on simulated data across the suite of five ML models.’ (Lines 311-314)

• 305: "Within epidemic transmission models, the SEIR model performed highest with randomly-sampled outbreak data" – highest compared to what? I was under the impression that you only used one epidemic model.

This has been addressed as described immediately above.

• 319: I'm not convinced that you "comprehensively investigated varying sampling types." It would be worth doing so, or if not, removing this statement.

This part of the sentence has now been removed (Line 321).

• 321: How did you identify that "diagnostic testing indication, sampling type, and the individual population tested are critical factors", and what are they critical for?

We have revised this section substantially in the Discussion, drawing on more specific elements of discussion, as below:

‘Our work highlights the need for these approaches to utilize data which are not conditioned on disease severity or other indicators, which impact the time between infection and specimen collection. For example, the long-term care facility dataset used for inference in the study required that specimens were collected randomly and not based on symptoms.’ (Lines 324-328)

• 323: How did you establish that "model selection must be tailored to the epidemiological circumstances of testing"? And how should it be tailored to the circumstances?

We agree with reviewer that this was not firmly established by our analysis, and have now removed this sentence. Rather, we now emphasize several variables to consider to help inform model selection, as immediately above and as now described below:

‘The rate of sampling is also important to consider with regards to accuracy of constructed incidence. Results from the long-term-care facility indicate this modeling approach demonstrated a slight difference in incidence peak timing and amplitude across the horizons. This is likely explained due to the lag time between onset of the infectious period and reporting given that site-wide facility testing was performed at set time periods rather than on a daily basis, and represents a pragmatic approach to real-world settings. This supports utility of this approach for other similar settings such as long-term care or assisted living or community-living facility outbreak investigations such as shelters, or within small hospital systems. In contrast, the novel application of machine learning approaches described in this study performed the best with large datasets (such as >1,000 COVID-19 positive cases), making this the approach of choice for large population settings such as at the province, state or large hospital network system level. Indeed, machine learning models can offer greater flexibility by incorporating different summary statistics and other data as features, fully harnessing the potential of larger datasets.’ (Lines 329-341)

• 325: "SEIR modeling was only suitable for random sampling." – I don't believe you have established this. If I understand correctly, you fit this model to two datasets, one of which was randomly sampled (the care home dataset). Visually, the model appeared to fit the population (non-randomly sampled) dataset better.

We agree with this feedback, and have revised the Discussion to better capture the study results and implications:

‘A consideration that we highlight in this study is the requirement of a sufficiently random sampling scheme to accurately estimate incidence from Ct value.’ (Lines 375-377)

• 327: "difference in peak timing" – difference from what? Do you mean across the horizons?

This has been added for clarity:

‘Results from the long-term-care facility indicate this modeling approach demonstrated a slight difference in incidence peak timing and amplitude across the horizons. This is likely explained due to the lag time between onset of the infectious period and reporting given that site-wide facility testing was performed at set time periods rather than on a daily basis, and represents a pragmatic approach to real-world settings.’ (Lines 330-334)

• 333-336: This sentence is misleadingly written. It suggests that the ML models work better with large datasets than the SEIR model – but you did not attempt to fit the SEIR and ML models to the same dataset. Instead I think you showed that the ML models show improved fit when a larger sample of the Ct values were used to train the model.

We have revised this sentence accordingly:

‘In contrast, the novel application of machine learning approaches described in this study showed improved fit with large datasets (such as >1,000 COVID-19 positive cases), making this suitable for large population settings such as at the province, state or large hospital network system level.’ (Lines 336-339)

• 340: "all approaches described in this study could predict future trends within a one- to four-week timeframe" – I'm not sure you established this. For instance, I did not see this for any of the ML models, and it was not really quantified for the SEIR model.

We thank the reviewer for this, and have toned down this statement to more accurately reflect the results produced by this study:

‘Importantly, the approaches described in this study demonstrated utility for timely prediction of SARS-CoV-2 transmission dynamics that could be harnessed to help inform future outbreak resource allocation and decision-making. For example, use of these models could be used to support decision-making across several settings, including hospitals, long-term care facilities, public health departments and others, to help inform planning of resource allocation, vaccination efforts, and isolation practices. More specifically, this approach lays the groundwork for a sentinel surveillance monitoring strategy that could be automated and alert appropriate authorities at pre-determined signals of predicted incidence changes, and may be expanded to other infections for which testing is widespread and predictive tools are needed.’ (Lines 343-351)

• 375: I believe you only fitted one model to the care-home data, so this sentence is misleading (or perhaps just a typo). Moreover, it is not clear to me why this approach is classed as validation, while the others are not, as it seems like in all cases you fit the models independently to the relevant data.

We have corrected this sentence to reflect the single model in the long-term care outbreak. In response to this feedback, we have better aligned use of the word validation throughout the manuscript as well:

‘Additional strengths of this study also include the independent assessment in a long-term care facility outbreak to validate the previously-published model (3).’ (Lines 371-373)

• More discussion of limitations is needed.

We thank the reviewer for this feedback, and have now substantially revised this section as follows:

‘However, there are several limitations. Firstly, due to limitations in the models used and insufficient numbers of individuals tested in the asymptomatic setting, we could not directly compare the performance of the two modelling strategies. However, we comprehensively investigated each approach and emphasized key performance metrics for each. Further work will be required to generate a head-to-head comparison, and fully characterize the relative advantages and disadvantages of each. Secondly, the methodology used assumed random sampling which is challenging to confirm. Indeed, testing practices were modified following clinical and public health guidance of the province, and may have led to bias in sampling. Restriction of the study population to the asymptomatic subgroup consisting of travelers and occupational health testing led to greater confidence in the employed sampling strategy tested and the validity of this assumption; nonetheless, there remains a need to develop a robust set of modelling approaches that can be leveraged across the broad variation of real-world sampling strategies. Thirdly, even though the long-term care environment provides more a consistent testing environment, it tends to be a highly vaccinated population which may potentially introduce bias. In addition to above, complementary approaches that may be better suited for analysis of small populations including stochastic modelling should be investigated for future work. Fourthly, this study did not leverage VoC or vaccination data, which are important potential confounders and for which the impact on viral load dynamics should be explored. Similarly, the models did not account for COVID-19-specific features including quarantine and social distancing, nor for wider applicability of the methods including long COVID and more recent VoCs, which may be relevant for future work in this field. Finally, this study aggregated Ct-level data across more than one assay from a single gene target, which may not adequately capture intra- and inter-assay variation or other gene target experience.’ (Lines 379-401)

Reviewer #2 (Recommendations for the authors):This study investigates how machine learning methods using cycle threshold (Ct) values – the semi-quantitative measurement output from most RT-qPCR tests – from routine laboratory testing for SARS-COV-2 can accurately estimate SARS-CoV-2 epidemic trends. The authors obtained an impressive province-level dataset of SARS-CoV-2 RT-qPCR testing on nasopharyngeal swabs collected during the BA.1 Omicron wave in British Columbia, Canada, and linked this to a vaccination status database to stratify Ct values by date, vaccination status and variant of concern (VoC). However, it is not clear how the vaccination and VoC data are used in the machine learning analyses, as the authors only describe the predictive performance of the machine learning models applied to simulated data. The study also applies an existing epidemiological modelling approach to estimate the incidence curve from a well-studied outbreak in a long-term care facility using Ct values, finding that with multiple random samples, the underlying incidence curve for the outbreak could be estimated accurately. The epidemiological model is also applied to the province-level BC data to estimate a plausible infection incidence curve.Overall, I do not think the authors have sufficiently achieved their aims, as the most robust results are largely an off-the-shelf application of an existing method to new datasets. There are also several conclusions that are not supported or contradicted by the results, often with no corresponding analyses presented. For example, the authors describe data on vaccination status and variants of concern in detail and mention that these data are included in the machine learning models (L371). However, the only results regarding the performance of the machine learning models use simulated data, and thus it is not clear how and where these additional real data are included. Another example is that the authors describe early on the focus on data from a restricted time period where most tests are expected to be in asymptomatic individuals (L105). However, on L231 the authors state that simulated data was used for the analysis due to testing guidelines tailored to symptomatic individuals. These and other inconsistencies weaken the claims of the paper.Strengths:The two main data sources for this study – the Provincial Health Laboratory Viewer and Reporter (PLOVER) and the Provincial Immunization Registry (PIR) dataset – have large sample sizes and if used as described would be fantastic datasets for testing these methods. The application of multiple machine learning and epidemiological modelling approaches to the same task of estimating the reproductive number and/or incidence using Ct values would be interesting if the accuracy of these various approaches were compared more robustly. I think this could be an important contribution. If the task of infectious disease surveillance is to accurately predict epidemic trends, then it is important to use state-of-the-art prediction methods like those used here. I also commend the authors for including the modelling code and simulated data on GitHub, which appears to be sufficiently documented.Weaknesses:The main weakness of this study is that many of the conclusions and described analyses are not presented in the manuscript as described above. It is often unclear which findings are related to real-world data and which are based purely on simulated data. For example, the reader is given the impression that the ML models are trained on simulated data and then tested on the real-world data to estimate the effective reproductive number for BA.1 Omicron in British Columbia during Nov/Dec 2021 (see L44, the Results section of the abstract). However, this analysis is never shown. The authors also claim to be addressing limitations of previous methods which do not account for changes in viral load kinetics due to vaccination and different variants, and non-random sampling strategies. However, none of the analyses this study presents account for these factors either. There are other studies on this topic that also consider non-random sampling, for example, Lin et al. Nature Communications 2021 https://doi.org/10.1038/s41467-022-28812-9.Another weakness of the study is the lack of information with which to appraise the performance of the machine learning approaches for predicting the effective reproductive number. First, the only figures on predictive performance are the MSE results in Figure 3, which do not demonstrate how well the approach reconstructs the incidence curve. Second, as far as I can tell the ML approaches assume complete reporting of the Ct distribution statistics each day, which would not be feasible in reality.The presented figures are hard to read, largely because the resolution is too low to read the smaller text (this can be corrected). Furthermore, the presented posterior estimates for the viral kinetics parameters in Figures 4 and 5 are impossible to interpret without further explanation/context unless one is already completely familiar with this model. The authors state that the values for these parameters were modified to fit the Omicron period, but what these modifications were and the rationale behind them are not explained.I also find the inclusion of all the Ct distribution moments and the interpretation of their variable importance questionable. These metrics are highly correlated (e.g., mean and median), and thus I would expect there to be some more care in choosing which statistics to include. I also wonder if the importance of variance (as per the SHAP plots) may be because the methods are indirectly picking up sample size (i.e., number of positive tests) from variance – when there are very few positives then the empirical variance of the Ct distribution may be very low, but when there are many positives then the full spectrum of Cts may be represented.Finally, while a semantic point, the study purports to investigate *prediction* of epidemic trends. The term prediction in infectious disease modelling is usually reserved for analyses seeking to extrapolate trajectories into the future. All the analyses presented here deal with nowcasting or hindcasting – estimating epidemic trends that have occurred up to the latest sampling date. Thus, I would not consider the methods here to be tackling a prediction task.Impact on the field:While the aims of this study are important to the field of infectious disease modelling and surveillance, the analyses would need substantial revision and extension to provide new insights beyond those already shown by previous work. The use of machine learning methods applied to Ct value distributions to predict infection incidence is an interesting topic, and understanding how time-varying vaccination coverage and circulation of variants with different viral kinetics impact these Ct distributions is interesting. However, this study is currently inadequate to address those questions.

We thank the reviewer for this very insightful feedback, and note that these comments align with reviewer #1. We have incorporated this feedback throughout our revised manuscript.

I think this is a worthwhile study and the province-level BC dataset sounds fantastic. The availability of code is good, though the data are not provided (only a link to where they can be requested). I think the analyses and presentation would need to be substantially overhauled to support the claims made here. In addition to the broad points discussed above, some suggestions for improving the paper and specific areas which need clarifying are:1. The ML results feel very incomplete and confusing. First, the mentioned analyses considering vaccination and VoC data should be presented and interpreted. Otherwise, I am not sure why those data are described and presented.

We thank the reviewer for these comments, and note that this was also brought up by Reviewer 1. As such, we have substantially revised the Methods to better explain the SEIR and ML approaches.

Given vaccination and VoC data were not explicitly accounted for in the current study, we have now removed emphasis on these in the Introduction and Discussion, and highlighted this as a limitation:

‘Fourthly, this study did not leverage VoC or vaccination data, which are important potential confounders and for which the impact on viral load dynamics should be explored.’ (Lines 394-396)

We have kept the VoC and vaccination data in Table 1 to provide context of the population studied.

Second, it needs to be much clearer where the ML methods are using simulated or real data. The idea of training the ML model on simulated data and then applying it to the real dataset is interesting, but I did not see any of those results.

This has now been addressed in the revised manuscript, as described earlier in this document (Lines 215-237).

Third, I would suggest showing the epidemic curves or Rt estimates outputted from the ML models rather than just the overall MSE summary plots.

This has now been added to the revised Figure 5.

2. To claim that the study is investigating the prediction of epidemic trends rather than just reconstruction would require a formal assessment of prediction at future time horizons. On L340 you state that "all approaches described in this study could predict future trends within a one to four-week timeframe", but I see no results to support this claim.

We thank the reviewer for this – which is aligned with reviewer #1 – and have toned down this statement to more accurately reflect the results produced by this study:

‘Importantly, the approaches described in this study demonstrated utility for timely prediction of SARS-CoV-2 transmission dynamics that could be harnessed to help inform future outbreak resource allocation and decision-making. For example, use of these models could be used to support decision-making across several settings, including hospitals, long-term care facilities, public health departments and others, to help inform planning of resource allocation, vaccination efforts, and isolation practices. More specifically, this approach lays the groundwork for a sentinel surveillance monitoring strategy that could be automated and alert appropriate authorities at pre-determined signals of predicted incidence changes, and may be expanded to other infections for which testing is widespread and predictive tools are needed.’ (Lines 344-352)

3. It would be useful to see results on how well the ML models predicted (rather than just nowcasted/hindcasted) the real and simulated epidemics at different time horizons, sampling frequencies, and under different epidemic patterns rather than just simple SIR-like shapes. The results here show that the epidemiological model from virosolver performs quite well for the datasets used here. This will not be the case in most settings for the reasons you discuss in the introduction, and thus understanding where the mechanism-agnostic ML methods can outperform the epidemiological model would be useful.

We thank the reviewer for this feedback. The simulated Ct data were generated across different sizes and a fixed sample period, whereas the ML models are assessed on out-of-sample data present within the same simulated Ct dataset to mimic real-world data and the scores for the models are present in the boxplots in Figure 5. Unfortunately, due to the insufficient number of pseudo-random samples, these models could not be assessed on real-world data in the current study. We have added clearer description of this in the methods, and highlighted this limitation in the discussion.

‘A separate real-world data analysis by ML was planned for the study; however, due to the insufficient number of randomly-tested samples with which to conduct the study with real-world data, this could not be performed and only simulated data were used.’ (Lines 218-220)

And

‘Firstly, due to limitations in the models used and insufficient numbers of individuals tested in the asymptomatic setting, we could not directly compare the performance of the two modelling strategies. However, we comprehensively investigated each approach and emphasized key performance metrics for each. Further work will be required to generate a head-to-head comparison, and fully characterize the relative advantages and disadvantages of each.’ (Lines 379-384)

4. The presentation of all the viral kinetics parameter posteriors is very odd (Figures 4 and 5), and I think would confuse most readers. I am very familiar with that work so understand what you are showing, but I think most readers would not. For example, the parameter names in Supplementary Table 6 are not interpretable without more context. Indeed, those parameters are largely "nuisance" parameters, and if the posterior estimates are biologically plausible and do not deviate too far from the priors, then they are not the main outputs of interest. It was also unclear to me what modifications to these parameters were made and why – were there some references on Omicron viral kinetics that were incorporated? The priors are also not shown here, only the upper and lower bounds. The columns "steps", "lower start" and "upper start" have no relevance to the reader and are specific to the virosolver code.

Figures 4 and 5 now correspond to Figures 4 and 3, respectively. We have now added information in the revised manuscript and legends to provide better context and facilitate interpretation, including:

Figure 3 legend:

‘Long-term care facility outbreak investigation modeling findings. A multiple-cross section SEIR model was fitted to the outbreak data (A), and showed a peak in incidence on the 12th day of the outbreak which preceded by two days the observed peak at the outbreak facility. The population included in this outbreak investigation was sampled at three pre-determined time points (dashed red lines). The Monte Carlo chain model-predicted incidence curve is represented by black lines, and was overlaid with the reported number of confirmed SARS-CoV-2-positive cases in this outbreak setting (yellow bars). Violin plots of the viral kinetic parameters for the SEIR model are also presented in the outbreak case study (B). The MCMC approach searches over the viral kinetics described above and is based on prior values (virosolver). Fit to detectable Ct distribution across time points 4, 12 and 19 are also presented in the outbreak study (C). These show the model fit (blue curve) overlayed with the frequency of Ct values (grey bars) and are a good indicator of the Ct distribution across the time points. The Ct values increase from day 12 to 19 as the epidemic declines.’

Figure 4 legend:

‘Overall population modeling findings. A multiple-cross section SEIR model was fitted to the overall population-level data (A), and showed an incidence peak from December 27 2021 to January 1 2022, which overlapped with the observed peak of reported cases in the province. The Markov chain Monte Carlo model-predicted incidence curve is represented (black lines), and was overlaid with the reported number of confirmed SARS-CoV-2-positive (yellow bars) cases. Violin plots of the viral kinetic parameters for the SEIR model are presented (B). Three unique time horizons were chosen of sizes 5, 6 and 7. The MCMC approach searches over the viral kinetic parameters presented above, and is based on prior values described separately (Supplemental Table 6). To align with the described Omicron viral kinetics, the incubation period was fixed and set at three days, and the infectious viral kinetic parameter was fixed. An upper bound of I0 was set at 0.100. The initial reproductive number (R0) increases across more horizons which in turn shifts the SEIR peak earlier. The fit to detectable cycle threshold distribution is presented over the largest horizon (C). The largest frequency (grey bars) of model fit lowest Ct values (blue curve) occur on days 14, 15 and 16 which represented the peak of the epidemic.’

In addition, we have revised Supplemental Table 6 (now Supplemental Table 5 in revised manuscript) as follows:

The parameter names are now explained in more detailThe columns ‘steps’, ‘lower start’ and ‘upper start’ have been removedThe variables that were modified relative to the original Hay *et al.* paper are now emphasized

Reviewer #3 (Recommendations for the authors):The authors study PCR (polymerase chain reaction) tests for COVID-19 in Canada, with investigation of the Ct (cycle threshold) on the tests. They use a variety of machine learning techniques to estimate the reproductive number (Rt) of COVID-19, based on received tests, and Ct. They validate their approach using "out of sample" data (i.e., holding out some of the data, and then predicting it.) This work is quite interesting, as Ct level data is under-studied. In addition, the problem is approached in a very pragmatic way in which many techniques are considered, and the results are well compared between methods. The methods considered are quite appropriate (for example, there is no need to have deep learning for such a square dataset.) Their results are strong.As the authors mention, usually PCR test results are reported as positive/negative, based on a threshold. In this work, Ct is the "input" and Rt is the "output". This work is an insight into rich data about COVID-19. The work also investigates specific outbreaks within BC, Canada, and also considers vaccine status, and shows that their Rt estimations are good. This work examines sophisticated datasets, so we must value the insights that are gained from such close study of rich data.In addition to validation on real data, this work also relies on simulations, and on the SEIR compartment model. This work could be improved by using compartment models more specific to COVID-19 (for example, with additional components for asymptomatic, or physical distancing), or more comparison with existing baselines. Further, estimation of Rt is not currently the most pressing concern with COVID-19 (can this work say anything about PACS, or MIS-C?)- The reliance on simulated data is too strong. For example, in Figure 3, the machine learning methods are compared. It appears that all of the data going into this main comparison between machine learning methods is based on simulated data. Also, I struggle to see what the y-axis is on this figure (it is labelled MSE, but there are no units given). "First modeling approach: machine learning. The fitted ML models were applied to out-of-sample Ct data from the simulated Ct values (Figure 3). With increasing sample sizes, the MSE across each model reduced by 82% showing … " the reference to this main figure does not tell me what is being measured on the y-axis.

We thank the reviewer for these comments, and have addressed these in the revised manuscript.

The Methods that underlie the ML analysis are now presented in more detail (Lines 215-237), including justification for the use of simulated data:

‘The second modeling approach was based on a collection of ML approaches for prediction of the reproductive number on simulated data, including Lasso (26), Random Forest (RF), Light Gradient Boosting Modeling (LGBM), eXtreme Gradient Boosted Modeling (XGBM) and CatBoost. A separate real-world data analysis by ML was planned for the study; however, due to the insufficient number of randomly-tested samples with which to conduct the study with real-world data, this could not be performed and only simulated data were used.’ (Lines 215-220)

For the Results section, we have revised this as follows:

‘The fitted ML models were applied to out-of-sample data from the simulated Ct values, and the predicted R_t_ of the five ML models were compared against the true simulated R_t_ across sample sizes 100, 1,000 and 10,000 (Figure 5A). With increasing sample size, the predicted R_t_ for the top performing model followed the true R_t_ more accurately (Figure 5A). Separately, the MSE was computed for each ML model comparing the predicted R_t_ with the true R_t_ (Figure 5B). Across all ML models, lower MSE (improved performance) was observed with increasing sample size (Figure 5B). The top performing model at sample size 100 was LGBM with a median MSE distribution of 0.14 (0.03). The top performing ML model for sample sizes 1,000 and 10,000 was Random Forest, with a median MSE distribution of 0.05 (0.007) and 0.02 (0.003), respectively. The MSE for the Random Forest model decreased by 82% from sample size 100 to 10,000 demonstrating improved performance of the moments of the Ct distribution to predict R_t_ on larger sample sizes.’ (Lines 289-300)

For the revised Figure 3B (now Figure 5B) legend:

(B) Boxplot of the Performance (MSE score) of all 5 models on 3 different sample sizes (N = 100, 1,000 and 10,000). Increasing sample sizes decreases the MSE resulting in a more accurate predictive model. Random Forest is the best model at higher sample sizes.

- Similar to above, while there is much reliance on simulated data, we do not see any details about how the data were simulated. "Ct data were generated to simulate a sufficiently large random sample of a population using the virosolver package". Some details about this method would be appropriate, or an indication of what parameter settings were used during the simulation, and why those parameters were used.

We thank the reviewer for this feedback, and note that this is consistent with the feedback from the other two reviewers. We have now added this information in the revised manuscript:

‘To simulate infection times and Ct values, a separate deterministic SEIR model adapted from the virosolver package was used for the machine learning approach. Full detail on the data simulation is included in the Supplemental Data, and all code used in the current study to enable reproduction is provided separately (https://github.com/Afraz496/Vital-E-paper). The simulation sample period was set at 140 calendar days to encompass a typical single SARS-CoV-2 wave. Ct values were generated to simulate a sufficiently large random sample of a population, and was applied on sample sizes of 100, 1,000 and 10,000 on a simulated population of 50,000 individuals.’ (Lines 220-227)

- Some parts of the paper rely on the SEIR model. This model is unrealistic for COVID-19, as quarantine, physical distancing, and asymptomatic cases are not considered. This work could be improved if a more sophisticated compartment model was used.

We agree with the reviewer that this model makes several assumptions which do not account for some unique characteristics of COVID-19. However, given one of the study objectives was to extend the validation work of the previously-published SEIR model, we elected to apply this model as is. We have now included this limitation in the Discussion:

‘Similarly, the models did not account for COVID-19-specific features including quarantine and social distancing, nor for wider applicability of the methods including long COVID and more recent VoCs, which may be relevant for future work in this field.’ (Lines 396-399)

- Regarding the VOCs, in Table 1 we see 28,580 samples without a known lineage. Can this be further resolved, if not why?

We have addressed this in the revised Figure 1 legend.

- I do struggle a bit to know the relevance of the results: While any work on COVID-19 may have been in preparation for a long time, I wonder if Rt is currently an object of interest? I'd consider long-COVID, further VOCs, host genetics, or future pandemics as a more pressing research question with respect to COVID-19.

We thank the reviewer for this insight, and agree that thought should be given to wider applicability and future research. As above, we have added this in the revised manuscript:

‘Similarly, the models did not account for COVID-19-specific features including quarantine and social distancing, nor for wider applicability of the methods including long COVID and more recent VoCs, which may be relevant for future work in this field.’ (Lines 397-400)

[Editors’ note: what follows is the authors’ response to the second round of review.]

The manuscript has been improved but there are some remaining issues that need to be addressed. In particular, new analysis is needed to address concerns raised in the first round of reviews and highlighted again in the review below. There are also still some claims that do not appear to be supported by the results as well as a number of areas where improvements in clarity and reporting of results are needed.Reviewer #1 (Recommendations for the authors):Thank you for the detailed response to my previous comments. I think the writing in this paper is now much improved and is less misleading and easier to follow than the previous version. Some of the claims are appropriately moderated in the discussion, and the methods are more complete and have a better structure.

We thank the reviewer for this feedback and the opportunity to improve the manuscript. We have carefully integrated the recommendations as described point-by-point below.

However, I still have some misgivings about this paper as it is. It's a little disappointing the authors did not undertake any new analyses based on the lengthy comments provided by the reviewers in the previous round. In particular, I think the paper would really be much stronger if the authors were able to compare the modelling approaches on the same datasets – I don't quite understand why this is not possible. A greater variety of simulated data could also be used, to test the models' capabilities. Additionally, there are still parts of the manuscript which appear to compare the SEIR and ML models (e.g. L303). There are also parts of the manuscript (e.g. L85) which may lead the reader to believe that this paper tests these approaches under biased testing regimes (which would be an interesting analysis), but I believe it only ever employs the methods under a regime random sampling – unless I am missing something.

We thank the reviewer for this feedback. We agree that the addition of new analyses are beneficial to strengthen the overall paper. As such, we have now performed a new direct comparison between the SEIR model and the ML model based on simulated Rt data using the virosolver package. The manuscript has been updated accordingly. Furthermore, we have revised Figure 3 to incorporate these new results for enhanced comparability.

We agree with the reviewed that head-to-head comparison of models on real-world data would have been ideal. We attempted this, but full analysis was not possible in the context of the current study due to the limited number of real-world samples in the real world. We have summarized this in the Methods discussion, and provided clearer emphasis on this limitation in the Discussion, as described below:

Methods:

‘We then applied the SEIR model in the current study to simulated data. To simulate infection times and Ct values, this SEIR model was adapted from the virosolver package (5). Full detail on the data simulation is included in the Supplemental Data, and all code used in the current study to enable reproduction is provided separately (https://github.com/BCCDC-DSI/Vital-E-paper). The simulation sample period was set at 140 calendar days to encompass a typical single SARS-CoV-2 wave. Ct values were generated to simulate a sufficiently large random sample of a population, and was applied on a sample size of 1,000 on a simulated population of 500,000 individuals. Based on this approach, the default viral kinetics including R_0_ and I_0_ from the virosolver package were used.’ (Lines 419-427)

‘Based on these analyses, we were able to produce head-to-head result comparison from the SEIR and the ML models comparison for simulated data, and to produce analysis on real-world data for the SEIR model only.’ (Lines 453-455)

‘A separate real-world data analysis by ML was planned for the study to ensure head-to-head comparison between the models; however, due to the insufficient number of randomly-tested samples with which to conduct the study with real-world data, performance was very limited with preliminary analyses and precluded further ML work based on real-world data.’ (Lines 432-435)

Discussion:

‘Firstly, due to the insufficient number of individuals tested in the asymptomatic setting to perform ML analysis, we could not directly compare the performance of the two modelling strategies for real-world data. However, we focused on the comparison between the two models for the simulated data, and investigated key performance metrics for ML analysis on which future research may build.’ (Lines 246-250)

Some more specific line-by-line comments• L80, 86-86: this still sounds like you are motivating the study in the same way as before – in terms of testing strategies and epidemiological settings. As the methods are the same as before, these things are still not established.

We thank the reviewer for this feedback. We have nuanced this section to better account for this:

‘Starting in December 2021 in British Columbia (BC), use of PCR testing was partially restricted in the context of roll-out of rapid antigen tests, limiting understanding of population trends. Complementary tools are needed to estimate incidence. This includes modeling approaches robust to varying testing guidelines, sample selection strategies and epidemiologic settings.’ (Lines 80-83)

• L89: Please write out these acronyms when they are first introduced.

We have now modified this accordingly.

‘In this study, we investigated the use of epidemic transmission modeling and machine learning (ML) including five models (Lasso, Light Gradient Boosting Machine [LGBM], Extreme Gradient Boosting [XGBoost], Categorical Boosting [CatBoost], Random Forest [RF]), based on Ct value distribution for SARS-CoV-2 incidence prediction in British Columbia, Canada during an Omicron-predominant period from November 2021 to January 2022.’ (Lines 84-89)

• The font size on a number of the figures is still very small – e.g. Figure 1A, 3A, 4A, …

We have now increased the font size as recommended for Figures 1A, 3 and 4.

• L189: I don't understand how it is possible that the model will fit real data but not simulated data. If the simulated data is of the same format as the real data, the process should be the same, surely? And as it's simulated data, it should be possible to generate it in the format you need?

We thank the reviewer for this comment, and have now addressed this by adding SEIR model analysis on simulated data as recommended. This is described in the response above regarding the edits made in the Methods, Results and Discussion sections.

• Relatedly, it's also still unclear to me why you can't fit the ML models to the province-level data – is it just a sample size issue? What happens when you try to fit the models?

We agree with the reviewer that adding this analysis would have been ideal. As described above, we attempted this but full analysis was not possible in the context of the current study due to the limited number of real-world samples in the real world. We have summarized this in the Methods discussion, and provided clearer emphasis on this limitation in the Discussion, as above below.

• L201-207: just to be clear, the model is fit to the Ct value distributions, but not the case data, correct?

This is correct, and we have modified this accordingly in the revised manuscript as follows:

‘Thus, the current study SEIR model was subsequently fitted to province-level Ct value distribution data from asymptomatic individuals using a MCMC framework.’ (Lines 403-404)

• L207: For the omicron adaptations, please could you give a reference for why you made these choices. It might also be useful to know what these parameters where in the original model.

We have now addressed this in the revised manuscript as suggested.

‘Based on earlier evidence, the initial time (t_0_) was fixed to 1 day, the incubation time was fixed to 3 days, and the infectious period was fixed to the default value of 4 days (5, 15). Fixed here implies that the viral kinetics were made static rather than dynamic by searching for the parameters via the MCMC framework, and these values were fixed due to sparsity of data. In addition, the model searched for I_0_, and the upper bound was set to 0.1 based on estimated provincial incidence during the timeline of the study.’ (Lines 412-417)

• Figure 2A: Recommend you present the lower and upper quartiles, rather than just the width of the IQR.

This has now been modified as recommended in Figure 2A.

• L264 – Rt is not measured in days

We have modified this accordingly. We have also modified Figure 3 legend to clarify this.

‘The Ct posterior predictive distribution across the three timepoints showed an increase in Ct values between outbreak days 12 to 19 as the number of cases waned (Figure 3C).’ (Lines 116-118)

• Figure 3A: It might be easier to see what is going on if you show mean and 95% credible interval, but not the sampled trajectories which obscure the 95% CrI

We have also edited the Figures and legends to describe that the blue ribbon represents the 95% credible interval.

• L260 and L274: what does it mean to say the incidence is within the 95% CrI? It looks like sometimes it is, but often it isn't. Relatedly, in the captions for Figures3 and 4, you do not describe what the blue ribbon represents (I am assuming this is the 95% CrI)

We have edited the revised manuscript accordingly, as described below. We have also edited the captions of both Figures to describe that the blue ribbon represents the 95% credible interval.

‘This model showed a peak in incidence on the 12th day of the outbreak which preceded by two days the observed peak at the outbreak facility (Figure 3A). The SEIR model demonstrated reasonable prediction with the real incidence falling within the 95% credible interval of the predicted MCMC chains. The model also accurately predicted the decline in cases by the 20th day of the outbreak (Figure 3A). A violin plot of the posterior samples shows a low outbreak incubation time with a median of 2.6 days, and a high initial Rt with a median of 9.5 days (Figure 3B).’ (Lines 110-116)

• In Figure 3A and 4A, could you also show the prediction interval (i.e. accounting for case reporting?) This might help to understand whether the estimated incidence matches the true incidence. Perhaps this is not possible with this model, in which case ignore me.Relatedly, you could then quantify the fit by reporting the coverage of the 95% prediction interval (proportion of incidence values which fall within the 95% range)

We agree with the reviewer that adding the prediction interval would be helpful. However, the sample in our study is built from a patient population where the denominator is unknown. As such, we could not realistically show the prediction interval; rather, we used this sample to create a general estimate of incidence for the entire population. We have added this limitation in the Discussion section.

‘Fifthly, as the sample in our study was built from a patient population of unknown denominator, we could not realistically show the prediction interval; rather, we used this sample to create a general estimate of incidence for the entire population.’ (Lines 268-270)

• L303: Again, you don't compare the models (SEIR vs ML), so I just don't think you can make claims like this.

We have edited this section as recommended.

‘Finally, the models presented relative advantages and disadvantages which may impact feasibility for implementation, and which are summarized separately (Table 2); however, this was limited by lack of ML analysis for the real-world data. Based on the results above, ML was found to be better suited for larger sample sizes and was flexible in design, but presented greater computational complexity for analysis.’ (Lines 160-164)

• L354: I may have missed it, but I don't believe you compared the Ct distribution of Omicron to other variants.

This statement was modified as recommended.

‘This study focused on a time period of Omicron (BA.1) predominance, and in the context of a sampled population with heterogeneous vaccination status, demonstrated accurate prediction of incidence based on overall Ct distribution and viral kinetics without incorporating individual-level vaccination status.’ (Lines 221-224)

• In several places in the results and discussion, I think you are overstating the success of these approaches. The SEIR appears to do quite poorly in the LTCF, and not outstandingly in the population-level data. The machine learning models similarly do okay, but not amazingly, even though the simulated context is likely to be much simpler than what would be encountered in real life, as it is based on a perfect epidemic curve.

We agree with the reviewer, and have carefully reviewed the entire manuscript and toned down the language to better capture these limitations. This includes the following examples:

Results:

‘Of the three horizon sizes tested, the best results were observed with a horizon of size 7 where the real incidence fell within the 95% credible interval of the predicted MCMC chains of the SEIR model (Figure 4A).’ (Lines 124-126)

‘The posterior predictive Ct distribution approximated the observed Ct distribution on each of the time horizons, supporting reasonable incidence projection independent of biases of testing guidance.’ (Lines 132-134)

Discussion:

‘In this study, we investigated the utility of two distinct modeling approaches based on cycle threshold values, epidemic transmission modeling and machine learning, for incidence prediction and R_t_ estimation, respectively. The SEIR model provided reasonable estimates for randomly-sampled outbreak data and at a wider level on provincial data for the asymptomatic subgroup, and Random Forest performed with favorable accuracy on simulated data across the suite of five ML models.’ (Lines 172-177)

Agree that this would be a valuable area to pursue further work in for the future. We have now highlighted this in the Discussion section.

‘Further work will be required to fully characterize the relative advantages and disadvantages of each, and to investigate the performance of ML models with more complicated Rt patterns.’ (Lines 250-252)

• Could you add confidence intervals to the Rt estimates from the ML model, e.g. by bootstrapping or some other approach?

We agree with the reviewer that this would have been helpful include. Unfortunately, our re-analysis was limited by computational requirements, and this addition would have to be included in future research efforts.

[Editors’ note: what follows is the authors’ response to the third round of review.]

The manuscript has been improved but there is one remaining issue from Reviewer #1 that needs to be addressed.Reviewer #1 (Recommendations for the authors):I thank the reviewers for carefully addressing all of my comments. I have just one remaining query, after which I am happy with this manuscript:One of my main comments on the previous round pertained to comparing the SEIR and ML models on simulated data. The authors state in their response that they do this in Figure 3; however, I can't see how figure 3 does this--it seems instead to be just the SEIR results fitted to the real world data? Or probably I am missing something! I think I need to compare Figures5 and 6? Please could the authors clarify where this comparison is shown.

We thank the reviewer for pointing this out. Figures 5 and 7 best showcase the comparison of test performance between the two models. The references in the main manuscript accurately referenced these figures and so have been kept as is, as below:

‘By comparing the ML (Figure 7) with the SEIR model (Figure 5) on the same simulated data, the SEIR showed better performance compared to all ML models. The SEIR model presented an MSE of 0.62% (95% CI, 0.60-0.64%) and the best performing ML model presented an MSE of 54% (95% CI, 39-83%) (Table 2).’ (Lines 405-408)